

# Deep Learning Methods for Flood Mapping: A Review of Existing Applications and Future Research Directions

Roberto Bentivoglio[1], Elvin Isufi[2], Sebastian Nicolaas Jonkman[3], and Riccardo Taormina[1]

[1]Department of Water Management, Faculty of Civil Engineering and Geosciences, Delft University of Technology
[2]Department of Intelligent Systems, Faculty of Electrical Engineering, Mathematics and Computer Science, Delft University of Technology
[3]Department of Hydraulic Engineering, Faculty of Civil Engineering and Geosciences, Delft University of Technology

**Correspondence:** Roberto Bentivoglio, r.bentivoglio@tudelft.nl

**Abstract.** Deep Learning techniques have been increasingly used in flood management to overcome the limitations of accurate, yet slow, numerical models, and to improve the results of traditional methods for flood mapping. In this paper, we review 58 recent publications to outline the state-of-the-art of the field, identify knowledge gaps, and propose future research directions. The review focuses on the type of deep learning models used for various flood mapping applications, the flood types considered, the spatial scale of the studied events, and the data used for model development. The results show that models based on convolutional layers are usually more accurate as they leverage inductive biases to better process the spatial characteristics of the flooding events. Models based on fully-connected layers, instead, provide accurate results when coupled with other statistical models. Deep learning models showed increased accuracy when compared to traditional approaches and increased speed when compared to numerical methods. While there exist several applications in flood susceptibility, inundation, and hazard mapping, more work is needed to understand how deep learning can assist real-time flood warning during an emergency, and how it can be employed to estimate flood risk. A major challenge lies in developing deep learning models that can generalize to unseen case studies. Furthermore, all reviewed models and their outputs, are deterministic, with limited considerations for uncertainties in outcomes and probabilistic predictions. The authors argue that these identified gaps can be addressed by exploiting recent fundamental advancements in deep learning or by taking inspiration from developments in other applied areas. Models based on graph neural networks and neural operators can work with arbitrarily structured data and thus should be capable of generalizing across different case studies and could account for complex interactions with the natural and built environment. Physics-based deep learning can be used to preserve the underlying physical equations resulting in more reliable speed-up alternatives for numerical models. Similarly, probabilistic models can be built by resorting to Deep Gaussian Processes or Bayesian neural networks.

## 1 Introduction

Flooding is one of the most dangerous and frequent natural hazards, accounting for significant human and economic losses every year (Jonkman and Vrijling, 2008). Because of the climate change effects, more frequent and intense extreme precipitations are expected to further increase the severity of this hazard (Masson-Delmotte et al., 2021). To mitigate the impact of floods



on human lives and properties, both preventive and emergency measures are required (European Union, 2007). Emergency

measures are operations carried out just before, during, or after a flooding event. In those cases, real-time knowledge of the extent of the flood and the areas in danger is needed to execute countermeasures (Lendering et al., 2016). Instead, preventive measures are operations aiming at reducing the possibility of a certain area being flooded. Those can be determined by maps that indicate the hazard of floods, i.e., the potential flood characteristics for an event.

There are three main flood maps used for dealing with such measures: (i) flood extent or *inundation* maps determine the

observed inundation extent, during or after the event, and are used for emergency measures; (ii) *susceptibility* maps provide a qualitative categorization of the flood hazard in an area, given its physical characteristics, and are used for preventive measures; (iii) flood *hazard* maps indicate the spatial distribution of variables that characterize the flood hazard of a specific event, such as flood depth and water extent, and are used for both emergency and preventive measures. Traditionally, inundation maps are obtained via remote sensing analysis (e.g., Lin et al., 2016), susceptibility maps with multi-criteria decision analysis

(MCDA) (e.g., Abdullah et al., 2021), and hazard maps with numerical methods (e.g., Dottori et al., 2021). Despite their wide usability, each method has its limitations. Remote sensing analysis for flood inundation requires manual or semi-automated procedures to improve the results and additional data such as land cover distribution (e.g., Manavalan, 2017). In addition, traditional models for flood inundation are not scalable to large amounts of data, as the ones currently produced by worldwide satellite missions. MCDA for flood susceptibility is simple and interpretable but its results are not accurate for complex phe-

nomena (Khosravi et al., 2020). Moreover, the weights assigned to each criterion are subjective and thus biased by the external choices. Numerical methods for flood hazard modeling are robust and effective, but fast and accurate flood simulations remain a challenge (Costabile et al., 2017). There exist several ways to improve the speed of the simulations, for example, through parallel computing (e.g., Zhang et al., 2014; Ming et al., 2020; Glenis et al., 2013), or simplified models (e.g., Zhao et al., 2021a; Sridharan et al., 2021). However, parallel computing has high computational costs and simplified models are unable to

correctly reproduce rapidly-evolving flows such as in urban floods (Costabile et al., 2017) and dam breaks (Prestininzi, 2008). Moreover, numerical models have intrinsic limitations which depend on the discretization of the governing physical equations and physical domain.

To overcome these limitations, practitioners and developers have used data-driven models based on machine learning. Machine learning (ML) is a branch of artificial intelligence in which a model improves its performance, with respect to some class

of tasks, as the available data increases (Mitchell, 1997). Conventional ML techniques require specific feature engineering of raw data before its processing. Deep learning (DL) can, instead, automatically discover the representations needed for detection or classification in raw data (LeCun et al., 2015). Nonetheless, data must be carefully selected according to the task at hand. DL methods are representation-learning methods with multiple levels of representation, obtained by composing simple but non-linear modules that each transform the representation at one level (starting with the raw input) into a representation at

a higher and more abstract level (LeCun et al., 2015). The model can then learn hidden patterns in the data and, consequently, improve its performance. Both ML and DL models have been applied in the fields of hydraulics and flood analysis. Mosavi et al. (2018) examined ML models for the prediction of floods, in the short and long term. Sit et al. (2020) reviewed deep learning models for hydrology and water resources, focusing also on the hydrological modeling of floods. Zounemat-Kermani



et al. (2020) reviewed neurocomputing for surface water hydrology and hydraulics including some applications concerning
floods.

The existing reviews mainly focused on the temporal variability of floods, especially concerning rainfall-runoff modeling, covering only a few instances of flood mapping applications. But the spatial evolution of flood events is extremely important to determine affected areas, plan mitigation measures and inform response strategies. Yet, there are no comprehensive overviews and analyses of DL in flood mapping to facilitate flood researchers and practitioners. The aim of this review is thus to advance
the emerging field of DL-based flood mapping by surveying the state-of-the-art, identifying outstanding research gaps, and proposing fruitful research directions.

papers are analysed considering two main parallel yet intertwined directions. On the one hand, we focused on the flood management application, spatial scale of study, and type of flood. On the other hand, we examined the deep learning model, type of training data, and performance with respect to alternative methods. This strategy provides insights from a flood manage-
ment perspective and concurrently facilitates reflection on how to successfully apply DL models. For each category, we identify common patterns and deduce considerations based on the presented results, while also identifying papers presenting innovative approaches. Moreover, we compare against traditional methods to further validate the benefits of employing DL models. Finally, we highlight a series of current knowledge gaps and propose possible solutions to them, by drawing parallelisms with the recent advances in deep learning.

The remaining of this review is organized as follows. In Section 2, we present the background theory on both floods and deep learning. Then, in Section 3, we present the search methodology and discuss the results based on the reviewed papers. In Sections 4 and 5 we present the knowledge gaps and propose possible future research directions. Finally, conclusions are provided in Section 6.

## 2    Background

This section is divided in two parts: flood management and deep learning. In the first part, we present the categories in which we classify flood management, while in the latter we describe the main deep learning models used for flood mapping.

### 2.1    Flood management

Floods can be defined as an overflow of water in otherwise dry land. Hence, flood management is a very broad field of interest: wherever there is water, there is a certain probability of being affected by it. While there exist several categorizations of flood
management, we focus on types of floods, applications, and spatial scales.

#### 2.1.1    Types of floods

We can distinguish flooding depending on how, why, and when it occurs:

– **River floods** are caused by extensive precipitation over long periods, causing the river to overflow its banks, ultimately inundating the neighboring areas. This process is slow and can last for several days (Serinaldi et al., 2018).





– **Flash floods** are caused by short but intense rainfall or sudden melting of snow (Sikorska et al., 2015). They are rapid
        and intense floods, typical of mountain and steep catchments. Flash floods are usually coupled with other hazards such
        as debris flows (Destro et al., 2018) and landslides (Ávila et al., 2016).

     – **Coastal floods** are caused by extreme meteorological conditions, which increase the water level in large bodies of water,
        due to a combination of low atmospheric pressure and strong winds. They occur near oceans, seas, or large lakes and we
include in this category also tsunamis, although they are generated by geological phenomena such as earthquakes.

     – **Urban floods** are caused by the failure of drainage from a sewer system, due to extreme precipitation, resulting in the
        overflow of those pipes. Depending on the city position and topography, these floods can also be affected by all the other
        types of floods.

     – **Dam break and dike breach floods** are caused by the failure of flood protection structures, due to extreme flood events
or management issues. The uncertainty in if, where, and how a defence will fail further increases the unexpectedness of
        these phenomena.

     To simplify the categorization, we excluded pluvial flooding, i.e., floods caused by the failure of a drainage system due to
intensive precipitation. The underlying hypothesis is that pluvial floods can be addressed as urban floods in urban environments
or river floods if they also feature rainfall-driven river overflows.

**2.1.2   Flood mapping applications**

Since we focus on the spatial variability of floods, we distinguish among three types of mapping: flood susceptibility, flood
inundation, and flood hazard.

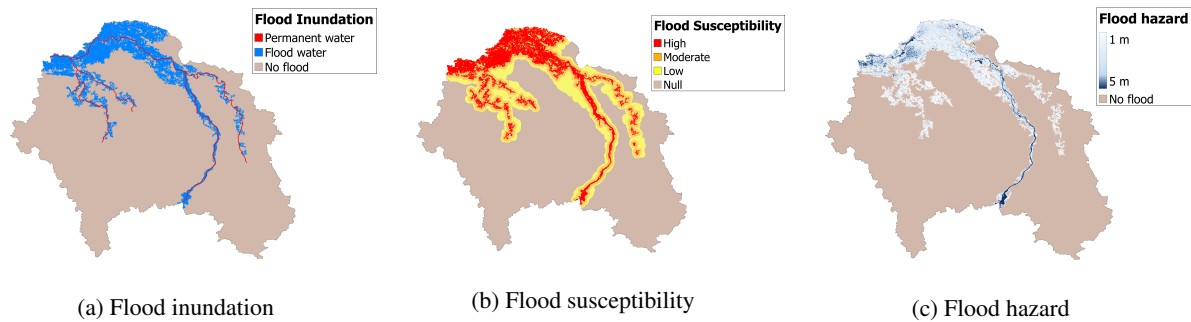

(a) Flood inundation          (b) Flood susceptibility          (c) Flood hazard

**Figure 1.** Examples of the types of flood maps analyzed for a representative area: (a) shows a possible flood inundation map; (b) a flood
susceptibility map; and (c) a flood hazard map, as defined in this paper.

     – **Flood inundation maps** determine the extent of a flood, during or after it has occurred (see Fig. 1a). Flood inundation
        maps represent flooded and non-flooded areas. This application is used for post-flood evacuation, protection planning,
and for damage assessment. These maps can then be used also as calibration data for other applications such as flood





susceptibility or flood hazard mapping. Flood images are obtained through remote-sensing techniques and processed by histogram-based models (e.g., Martinis et al., 2009; Manjusree et al., 2012), threshold models (e.g., Cian et al., 2018), and machine learning models (e.g., Hess et al., 1995; Ireland et al., 2015).

– **Flood susceptibility maps** determine the tendency to flooding of a study area based on its physical characteristics (see Fig. 1b). This measure is only qualitative and does not evaluate any flood variable. However, it can provide reliable information when no quantitative data is available and can be used to easily assess areas at risk at large scales. Flood susceptibility mapping is performed by considering topographical, geographical, and meteorological factors (such as altitude, slope, lithology, land use, and rainfall) and comparing their spatial distribution with past flood events. This is done with multivariate analysis (e.g., Tehrany et al., 2014; Youssef et al., 2016) and multi-criteria decision analysis (e.g., Kazakis et al., 2015; Mahmoud and Gan, 2018).

– **Flood hazard maps** measure the water depth and extent across a flooded area (see Fig. 1c). Hazard maps consider also different return periods of the floods and, thus, the probability of a certain event. The latter is determined through a statistical analysis based on the frequency and intensity of floods (Bobée and Rasmussen, 1995). We will refer to flood hazard also when the water depths are estimated, independently of the return periods. Flood hazard can also provide a measure of the flow velocities. Flood hazard maps are carried out by numerical models, which simulate flood events by discretizing the governing equations and the computational domain. We distinguish between one-dimensional (1D), two-dimensional (2D), and three-dimensional (3D) models with increasing complexity and, generally, accuracy (e.g., Horritt and Bates, 2002; Teng et al., 2017).

Flood damage and flood risk maps (de Moel et al., 2009) are other examples of mapping applications. However, they are not described in more details here as no related DL-based paper was found in the literature. Similarly, the review also excludes applications which do not result in maps, such as water level forecasts.

### 2.1.3 Spatial scale

The importance of flood processes and the resolution of the flood maps varies with their spatial scale. Following de Moel et al. (2015), we also distinguish between local, regional, national, and supra-national scales. The choice between scales is often subjective, but here follows a rational categorization:

– **Local** scale refers to small study areas, such as towns or a specific river stretch. If a measure of the study area is given, we consider it in this category if the area is smaller than $100 km^2$.

– **Regional** scale considers a specific province, watershed or large city. Study areas smaller than $100000 km^2$ belong to this scale.

– **National** scale refers to assessments of entire countries, for which consistent (national) data are present. To exclude small countries, the study area must be greater than $100000 km^2$.





– **Supra-national** scales concern assessments of an entire continent or the globe.

## 2.2 Deep learning methods

Deep learning studies how neural networks learn representations from data, through multiple levels of abstraction (LeCun
et al., 2015). A neural network is a non-linear compositional model formed by a hierarchical layering of parametric functions
that take an input variable $x$ and produce an estimate $\hat{y}$ of a target representation $y$ as $\hat{y} = f(x; \theta)$, where $\theta$ are the function's
parameters. The purpose of DL is then to calibrate those parameters to have the best fit between predicted output and real
output. The raw data $x$ are input to the neural network and the output of each layer serves as input for the following layer, until
the final layer, which coincides with the estimate $\hat{y}$. A neural network with $L$ layers can be expressed as

$$\hat{y} = f_L(\cdot; \theta_L) \circ f_{L-1}(\cdot; \theta_{L-1}) \circ \ldots \circ f_1(x; \theta_1),$$
$$x_\ell = f_\ell(x_{\ell-1}; \theta_\ell), \ \ for \ \ \ell = 1, \ldots, L,$$
$$\hat{y} \equiv x_L, \tag{1}$$

where $f_\ell(\cdot; \theta_\ell)$ is the function at layer $\ell$, $\circ$ represents the composition of functions, $\theta_\ell$ are the trainable parameters, and $\hat{y}_\ell$ is
the output layer $\ell$. In a network architecture, the layers between the input and the output layer are called hidden layers, since
their output is not shown. Estimating parameters $\theta_\ell$ is typically referred to as 'learning' and it is performed by minimizing a
loss function, through back-propagation (Rumelhart et al., 1986). Depending on the task, neural networks can be trained via
supervised and unsupervised learning. Since in flooding analysis DL has been mainly approached via supervised learning, we
focus on that learning process.

Supervised deep learning models identify a mapping from input to output, given a training set of input-output pairs. For
example, a training set for flood hazard mapping may comprise a flood's rainfall hyetograph as input $x$ and the corresponding
maximum flooded area as output $y$. Thus, the loss function $l(y, \hat{y})$ compares the real output $y$ with the predicted one $\hat{y}$. The
loss function is typically the quadratic loss for regression problems, where the data is continuous (e.g., water depth), or the
cross-entropy loss for classification problems, where the data is categorical (e.g., flooded and non-flooded areas). As training
data, we can have observations or simulations. Observational data are derived from remote sensing, flood inventory maps, and
measuring stations, while simulation data are derived from numerical solvers. Once a model is trained, its goodness of fit is
analyzed with a test set, composed of data that the model has not seen. If the model performs well for the test set, it is said to
*generalize* or extrapolate well. The ability to generalize is one of the most important properties of DL and becomes even more
important in high-dimensional inputs (Balestriero et al., 2021).

### 2.2.1 Multi-layer perceptron

Among the possible neural network layers, fully-connected ones are the most simple. In a fully-connected layer, the layer
propagation rule is given by:

$$x_\ell = f_\ell(x_{\ell-1}, \theta_\ell) = \sigma(W_\ell x_{\ell-1}), \tag{2}$$



(a) Multi-layer perceptron

(b) Encoder-decoder

(c) Convolutional neural network

(d) Convolution operation

(e) Recurrent neural network

**Figure 2.** Deep learning architectures: (a) a multi-layer perceptron (MLP) composed of a sequence of three fully-connected layers. Every layer is connected to the following one by weights, represented by directed arrows. The values of the input, hidden, and output layers are represented, respectively, by vectors $x_0$, $x_1$, and $\hat{y}$; (b) an MLP encoder-decoder. The input data $x_0$ is encoded into a lower dimensional layer $x_1$, and then decoded into the output $\hat{y}$. This structure is also applicable to convolutional and recurrent layers; (c) a convolutional neural network (CNN) composed of a convolutional layer, and a fully-connected layer. The green squares represent an input tensor, the orange squares represent hidden layers and the red parallelogram on the right represents the output layer. The small box $K_1$ represents the convolutional kernel described in Eq. 3. The final layer depends on the task; (d) visual explanation of how convolutional kernels work. Each element of the kernel is multiplied by its matching input value. Then, all values are summed to obtain the convolved output. This process is repeated across the whole input, as the kernel shifts along it. (e) a recurrent neural network (RNN) in compact form (left) and in the unfolded form (right). The iterative structure of the RNN (left) can be unfolded in time to show how hidden states influence the solution at each time step (right). The colouring scheme indicates for each architecture the input (green), the state (orange), and the output (red).



where $x_\ell$ is the output of the layer $\ell$, $\sigma(\cdot)$ is a point-wise nonlinearity (e.g., ReLU, $\sigma(x) = max\{0, x\}$, or Sigmoid, $\sigma(x) = \frac{1}{1+e^{-x}}$), $x_{\ell-1}$ is the input of the layer $\ell$, and the training parameter $W_\ell$ is a weight matrix. Multi-layer perceptrons (MLP) are composed by sequences of fully-connected layers (Fig. 2a). The expressivity of the network increases with the dimensions of the hidden layers, as shown in Fig. 2a. When the dimension of the hidden layers decreases and then increases, as shown in Fig. 2b, the architecture is called encoder-decoder (ED). The idea behind this architecture is that only certain latent representations of the input are useful to represent the output (e.g., Taormina and Galelli, 2018).

In fully-connected layers, the values of the parameters in $W$ are independent between them and there is no reuse of any of them. Thus, the number of learnable parameters is of the order of the input size, making fully-connected layers inappropriate for inputs of large dimensions. This issue is referred to as the 'curse of dimensionality' and implies that as the dimension of the input increases, the amount of training data needed to learn representations increases exponentially (LeCun et al., 2015).

To overcome the curse of dimensionality we need to exploit the structure in data. In flood analysis, data is usually structured: for example, neighboring pixels in raster data represent spatial proximity of nearby close elements, while discharge values in a hydrograph represent temporal proximities. Neural network layers can thus be defined in a way to exploit these data structures. These assumptions create what is known as an *inductive bias*, which imposes constraints on relationships and interactions among inputs in the learning process, thus prioritizing some solutions over others (Battaglia et al., 2018), as shown in Table 1. Inductive biases derive from the fundamental geometric principle of *symmetry* (Bronstein et al., 2021). The symmetry of a system is a transformation that leaves a certain property of said system unchanged. Symmetry results in invariance and equivariance properties. Invariance implies that transformations on the input features do not change the output (i.e., $f(g(x)) = f(x)$, $g(\cdot)$ being a generic transformation), while equivariance entails that transformations on the input features change the output via an equivalent transformation (i.e., $f(g(x)) = g'(f(x))$, $g'(\cdot)$ being a transformation equivalent to $g(\cdot)$). We explain the concept of invariance and equivariance with an example. Consider a picture with a flooded area in its top-left corner and one with the same flooded area shifted in the bottom-right corner. An invariant model would predict that there is a flooded area in both images, while an equivariant model would also reflect the change in positions of the flood, i.e., identify that the flood is in the top-left corner in one case and in the bottom-right corner in the other. In this case, invariance and equivariance are associated to a spatial translation, but the same principle applies to other transformations, such as temporal translation. Inductive biases thus lead to the reuse of parameters in different parts of the input of each layer. For instance, *convolutional* kernels can be used on images of different dimensions and *recurrent* layers can consider time series of variable length. Fully-connected layers, instead, cannot have such inductive bias capabilities. The main characteristics for each considered layer are synthesized in Table 1. The input data type and the inductive biases are described for each studied layer.

### 2.2.2 Convolutional neural network

Convolution is an operation for which every entry of an input matrix is replaced by a spatially weighted average of its neighbouring entries, as shown in Fig. 2d. The weights are defined by a matrix, called kernel, and are point-wise multiplied with the neighbouring entries. This procedure is then repeated, using the same kernel, for every entry in the input. Convolutional layers are a neural network layer that apply convolution on a input using trainable kernels, i.e., the kernels' weights are learned during





**Table 1.** Inductive biases and preferred types of data for different neural network layers (adapted from Battaglia et al. (2018)).

| Layer | Data type | Inductive bias |
|---|---|---|
| Fully Connected | Unstructured data | - |
| Convolutional | Grid elements | Spatial equivariance |
| Recurrent | Sequences | Temporal equivariance |

optimization (LeCun et al., 1995). The propagation rule of layer $\ell$ of a convolutional layer is

$$x_{\ell+1} = \sigma(K_\ell * x_\ell), \tag{3}$$

where $K_\ell$ is the kernel function for the $\ell^{th}$ layer and $*$ is the convolution operator. Convolutional layers are mostly applied to images i.e., two dimensional spatial grids. For such inputs the kernel is a 2D matrix. Convolutional layers have an inductive
bias of translational equivariance, which reflects the idea that spatially close grid elements influence each other. This results in the reuse of the same kernel across the different input parts and it implies that it matters where a pattern or object are in an image and that the model should be able to recognize it. Convolutional layers thus perform feature extraction, identifying relevant characteristics in the input. Moreover, the reuse of parameters allows inductive learning over images of different sizes or resolutions. Differently from fully-connected layers, the number of parameters in a convolutional layer depends only on the
kernel size because of this parameter-sharing property (see Fig. 2c). Depending on the input dimensions, we distinguish 1D convolutional layer for vector inputs, such as a rainfall hyetograph, 2D convolutional layers for matrix inputs, such as a digital elevation model (DEM), and 3D convolutional layers for tensor inputs, such as stacked satellite images. Since 1D-convolution considers translation equivariance on vectors, the inductive bias is equivalent to temporal equivariance if the vector is a time series.

Convolutional neural networks (CNN) are composed of layers alternating convolution and pooling. Pooling operation replaces the output at a certain location with a summary statistic of the nearby features, thus reflecting translational invariance (Bronstein et al., 2021). They extract a single feature, such as the average or maximum value in a certain neighborhood of a point. Furthermore, pooling reduces the dimension of the input, speeding up computation. The final layers of a CNN are typically fully connected when dealing with classification or regression tasks. This layer allows to map the convolved embeddings
to the number of classes or to the regressed value, respectively. Instead, if the task is to perform image segmentation, i.e., classify specific parts of an image, the final layers are composed of de-convolutional layers which perform the inverse operation of convolutional layers, in an encoder-decoder structure. For details on convolutional layers and CNNs refer to Goodfellow et al. (2016).

### 2.2.3 Recurrent neural network

Recurrent layers are used for processing sequential data, such as time series (Rumelhart et al., 1986). A recurrent layer can be seen as a nonlinear state-space model expressing the output at time $t$, $y_t$, as a function of a former hidden state $h_t$ and input $x_t$.





The basic formulation for a recurrent layer is

$$h_t = \sigma(W h_{t-1} + U x_t),$$
$$y_t = \sigma(V h_t), \tag{4}$$

where $U$, $V$, and $W$ are trainable weight matrices. As it follows from (4), the hidden state encodes the temporal memory of previous time instances while the output mapping is instantaneous. These matrices are shared across time allowing the recurrent layer to exploit temporal proximities of sequential data, irrespectively of their position. This is for instance the case of discharge hydrographs (e.g., Zhou et al., 2021). Because there is an inductive bias in temporal sequences, they allow us to reuse parameters without affecting the performance.

Recurrent neural networks (RNN) are neural networks composed of recurrent layers. The iterative structure of the RNNs can be unfolded in time to show how hidden states influence the output at each time step (Fig. 2e). However, the basic recurrent layer in (4) suffers from the problem of vanishing and exploding gradients (Hochreiter and Schmidhuber, 1997). This occurs due to the iterative use of the same layer which causes the weights to multiply several times when back-propagating the error, ultimately leading to "vanishing" gradients if the weights are small and "exploding" gradients if the weights are large. This 245    constrains then the temporal memory of these networks and limits their capability to extract long-term dependencies between the past inputs and the current output.

This problem is typically solved via the use of Long Short-Term Memory (LSTM) layers (Hochreiter and Schmidhuber, 1997). This variation of recurrent layers also improves the hidden state mechanism allowing to "remember" well even information which is temporally distant. Another common variation is the Gated Recurrent Unit (GRU) (Cho et al., 2014a), which 250    achieves comparable results with the LSTM architecture while using a simpler formulation. Same as for fully-connected and convolutional layers, recurrent layers can be used in encoder-decoder architectures. This structure can be composed of an RNN which generates a latent representation, followed by another RNN that decodes it (e.g., Cho et al., 2014b).

The most successful applications of RNNs for flood management regard tasks related to sequences and time-series analysis, such as rainfall-runoff modeling (e.g., Kratzert et al., 2019a). While RNNs are preferred over 1D-CNNs, recently the latter 255    started gaining momentum for some time-series learning tasks (e.g., Oord et al., 2016).

## 3 Review

### 3.1 Methodology

Papers were retrieved from the Scopus database by combining the keywords "deep learning" or "neural network" with "flood" or "flooding". The 3,338 publications obtained were then filtered to include only journal papers from January 2010 until 260    December 2021, in the areas of *engineering*, *environmental science*, and *earth and planetary sciences*. From this reduced list of 1308 papers, we considered two major refining criteria: i) the papers should be based on the deep learning models presented in Section 2.2, and ii) the applications must address the spatial variability of floods (i.e., not focusing only on the temporal aspects of flood analysis). This procedure resulted in 46 reviewable papers. This list was finally extended via a snowball





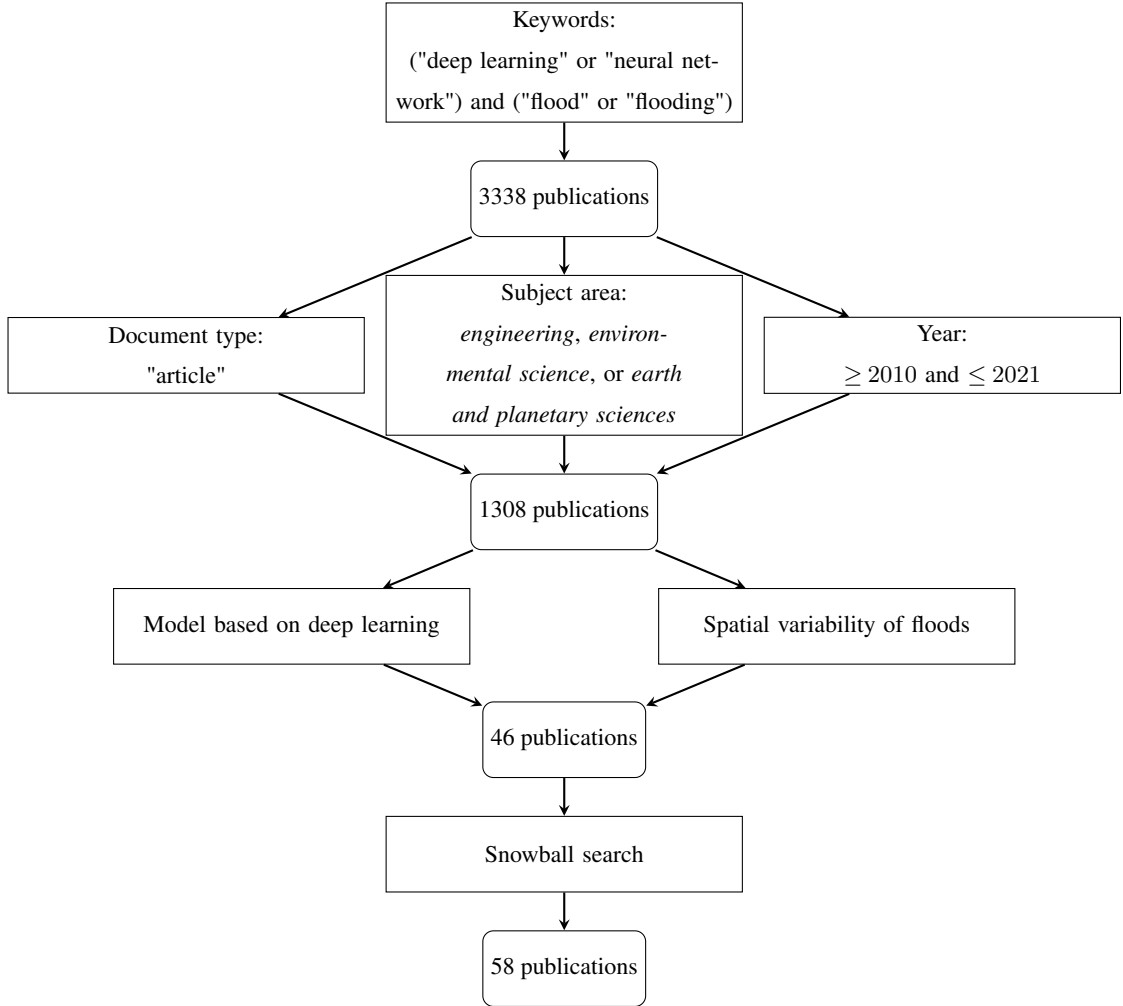

**Figure 3.** Flowchart of the methodology applied for the papers selection.

search that considered cited and citing works, ultimately leading to 58 eligible documents (Fig. 3). We find that the described

methodology selected a representative subset for producing a thorough review of recent advances and developments in this field.

     The selected papers are listed in Table 2 which reports major details including the flood mapping application, the type of flood, the DL model, and the spatial scale. General findings related to these three criteria are first presented in Section 3.2. Specific findings for each application are then presented in Sections 3.3 (flood inundation), 3.4 (flood susceptibility), and 3.5

(flood hazard). These specific sections provide a more in-depth discussion on the deep learning models employed, with a focus on the architecture, the input and output data, and the performance assessment.



Table 2: Deep learning applications for flood mapping. References are classified in terms of flood mapping application, type of flood, deep learning (DL) model, training data, and spatial scale.

| Application | Flood Type | DL Model | Training Data | Spatial Scale | Reference(s) |
|---|---|---|---|---|---|
| Inundation | River | MLP | Observations | Regional | Li et al. (2016a, 2015) |
| | | CNN | Observations | Local | Gebrehiwot et al. (2019); Nogueira et al. (2017); Hou et al. (2021); Ichim and Popescu (2020); Hashemi-Beni and Gebrehiwot (2021); Wieland and Martinis (2019) |
| | | | | Regional | Sarker et al. (2019); Kang et al. (2018); Nemni et al. (2020); Isikdogan et al. (2017) |
| | Urban | MLP | Observations | Local | Amini (2010) |
| | | | | Regional | Li et al. (2016b) |
| | | CNN | Observations | Local | Peng et al. (2019) |
| | | RNN, CNN | Observations | Local | Dong et al. (2021) |
| | Coastal | CNN | Observations | Regional | Liu et al. (2019); Isikdogan et al. (2017) |
| | | | Simulations | Regional | Muñoz et al. (2021) |
| | Dam Break | MLP | Observations | Regional | Syifa et al. (2019) |
| Susceptibility | River | MLP | Observations | Regional | Jahangir et al. (2019); Khoirunisa et al. (2021); Ahmadlou et al. (2021); Popa et al. (2019); Kia et al. (2012); Ahmed et al. (2021); Chakrabortty et al. (2021b); Saeed et al. (2021) |
| | | CNN | Observations | Regional | Wang et al. (2020b) |
| | | | | National | Khosravi et al. (2020) |
| | | RNN | Observations | Regional | Fang et al. (2020a) |





|  |  |  |  |  |  |
|---|---|---|---|---|---|
| | Flash | MLP | Observations | Regional | Tien et al. (2020); Ngo et al. (2018); Popa et al. (2019); Costache et al. (2020); Chakrabortty et al. (2021a) |
| | | | | National | Kourgialas and Karatzas (2017) |
| | | CNN | Observations | Regional | Panahi et al. (2021); Liu et al. (2021) |
| | Urban | MLP | Observations | Local | Darabi et al. (2021) |
| | | | | Regional | Kalantar et al. (2021) |
| | | CNN | Observations | Local | Zhao et al. (2021b, 2020b) |
| | | | | Regional | Lei et al. (2021) |
| Hazard | River | MLP | Simulations | Local | Chu et al. (2020); Huang et al. (2021a); Xie et al. (2021); Lin et al. (2020b, a); Jacquier et al. (2021) |
| | | CNN | Simulations | Local | Kabir et al. (2020); Hosseiny (2021) |
| | | RNN | Simulations | Regional | Zhou et al. (2021); Kao et al. (2021) |
| | Flash | CNN | Simulations | Local | Yokoya et al. (2020) |
| | Urban | MLP | Simulations | Local | Berkhahn et al. (2019); Chang et al. (2010) |
| | | CNN | Simulations | Regional | Guo et al. (2021); Löwe et al. (2021) |
| | Coastal | RNN | Simulations | Local | Hu et al. (2019) |
| | Dam Break | MLP | Simulations | Local | Jacquier et al. (2021) |

MLP=Multi-layer Perceptron; CNN=Convolutional Neural Network; RNN=Recurrent Neural Network

## 3.2 General findings

### 3.2.1 Flood mapping applications

Fig. 4 shows the distribution of papers for each of the applications considered: flood inundation, flood susceptibility, and flood

hazard. The research community has dedicated efforts to investigate each type of application, although flood inundation and susceptibility have received the most attention. While papers on flood inundation are more evenly distributed across years, applications for flood susceptibility and, especially, flood hazard are increasing in the last few years. Similar to what was observed in related fields such as hydrology (e.g., Sit et al., 2020), a strong surge in DL publications for spatial flood analysis is witnessed between 2018 and 2019. These years identify a turning point for AI in earth system sciences driven by the adoption





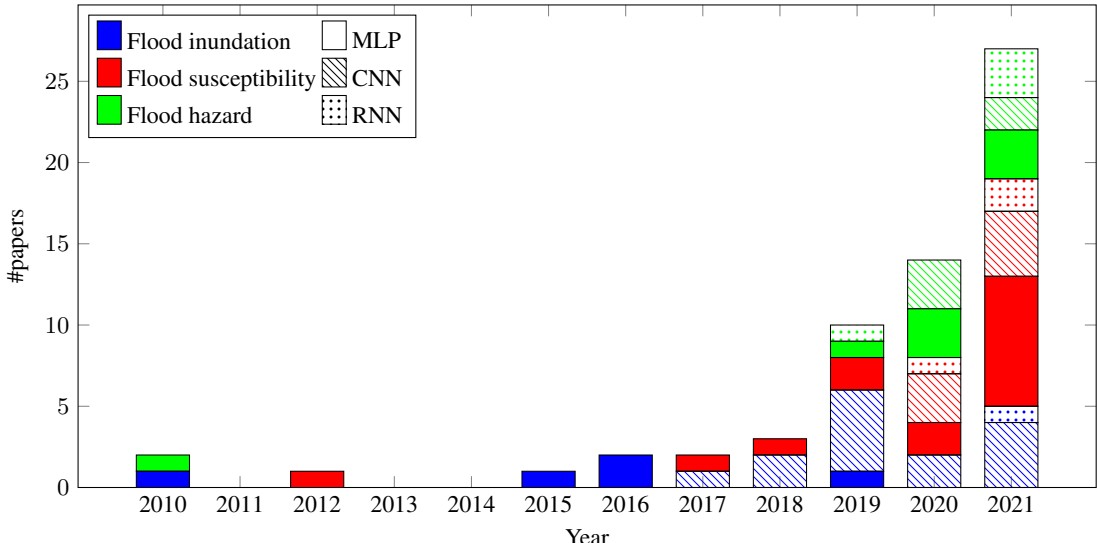

**Figure 4.** Publications by year, type of application and type of DL model. The increasing trend of the last five years has been mostly driven by the applications in flood susceptibility and flood hazard.

of CNN (striped patterns in Fig. 4) and RNN (dotted patterns) in lieu of traditional MLP models. The late use of convolutional and recurrent models is motivated by their recent popularization and development, along with a rise in awareness of the ML advancements, contrary to fully-connected layers, that have a longer application history.

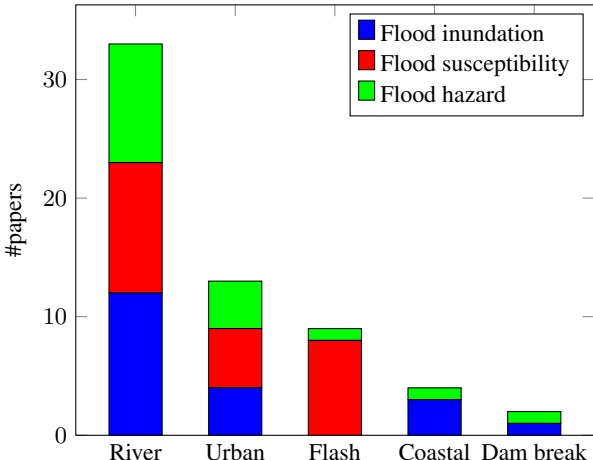

**Figure 5.** Distribution of the types of floods per flood application in the reviewed papers. River and urban floods are the most common, while flash and coastal floods have fewer occurrences.





### 3.2.2 Flood types

Fig. 5 shows the types of flood analyzed with respect to each application. River floods are the most common, with many
applications in inundation and hazard mapping. This is probably because, for historical reasons, most cities in the world are
built close to rivers (Kummu et al., 2011). The scientific community has dedicated significant efforts to exploring the potential
of DL for urban flooding. This is difficult to model because of the complex topography and the presence of a drainage system
whose dynamics need to be coupled with the overland flood (Löwe et al., 2021). Almost all papers analyzing flash floods
described flood susceptibility mapping applications. This is expected due to the short duration and the contingent nature of
these phenomena, which limit remote sensing imaging and numerical simulations used in flood inundation and flood hazard
mapping, respectively. Despite the importance of coastal flooding (Neumann et al., 2015), only a few papers report the use of
DL for coastal flooding. While other works are available in the literature (Lütjens et al., 2020, 2021; Bowes et al., 2021), they
were not considered since the employed DL models were not trained via supervised learning. Some of these works will be
discussed in Section 5. Dam break floods are the least analyzed type, possibly because of their relatively rare occurrence and
complexity.

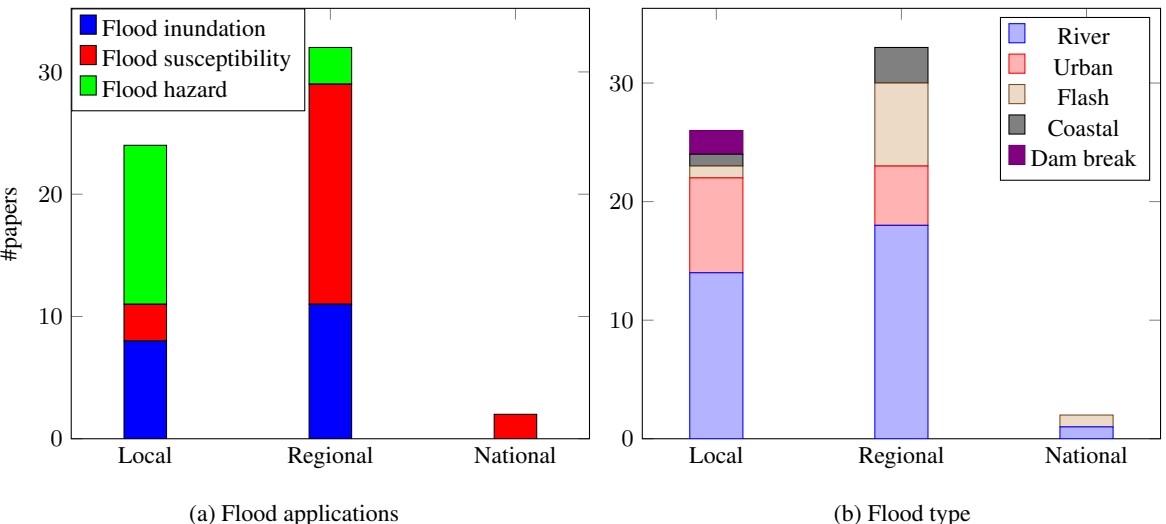

(a) Flood applications  (b) Flood type

**Figure 6.** Distribution of the spatial scale per (a) flood application and (b) type of flood in the reviewed papers. Local and regional scales are
the most used.

### 3.2.3 Spatial scale

As shown in Fig. 6a, most applications consider local and regional scales. Local scale refers to towns (e.g., Darabi et al.,
2021; Berkhahn et al., 2019), small catchments (e.g., Lin et al., 2020a; Kabir et al., 2020) or river reaches (e.g., Chu et al.,
2020; Gebrehiwot et al., 2019). As such, they are mostly referred to urban and river floods. The cases sizes vary from very
small ones, $165m^2$ (Hou et al., 2021), to small towns up to $100km^2$ (Lin et al., 2020a). Regional scale models consider a





catchment (e.g., Popa et al., 2019), a province (e.g., Wang et al., 2020b) or large cities (e.g., Löwe et al., 2021; Kalantar et al., 2021). Most works focus on river floods, while some study flash, urban, and coastal floods. National scale models refer to the assessments of entire countries, with only two papers concerning such scales, respectively for Iran and Greece (Khosravi et al., 2020; Kourgialas and Karatzas, 2017). Nemni et al. (2020) and Sarker et al. (2019) consider several study areas across

Africa and Asia, and Australia, respectively, but since the size of each area were smaller than $100000km^2$ they were marked as regional scale models. They also do not fit within the national scale classification since they do not encompass whole nations. Supra-national scale models assessing the entire globe or a continent have not been studied yet with deep learning models. This seems unexpected since ML techniques have already been employed at global scales, outperforming traditional techniques, for example in the estimation of design floods along river networks (e.g., Zhao et al., 2020a). Since DL models have been shown

to outperform ML models, as later outlined in this review, more models should be used at those scales in future studies.

### 3.2.4   DL architecture

Fig. 4 reports the architecture used for each application, showing that DL models are mainly based on fully connected and convolutional layers.

     MLP networks are widely used due to their flexibility and ease of implementation. However, they are usually coupled

with other techniques to reach satisfactory performances. Stochastic optimization techniques, such as genetic algorithm, firefly algorithm, and particle swarm optimization were combined with MLPs to search the optimal model's parameters (e.g., Li et al., 2015; Ngo et al., 2018; Kalantar et al., 2021). Multi-criteria decision analysis models, such as frequency ratio and analytical hierarchy process, were also coupled with MLPs to adjust the weights of each input in flood susceptibility (e.g., Kourgialas and Karatzas, 2017; Costache et al., 2020; Popa et al., 2019). Furthermore, k-means clustering was used to categorize the dataset

in classes, to account for different topographical conditions; then, for each class, a MLP was trained (e.g., Chang et al., 2010; Huang et al., 2021a). Combining MLPs with such methods partly compensates the lack of inductive biases, however, this lack blocks the model from employing existing structures in the data, ultimately limiting their usability. Since flooding phenomena have spatial and temporal structures, we expect MLPs to become progressively less used in this field, as hinted by the trend in Fig. 4.

CNNs are best suited for processing raster files and images, thanks to their spatial inductive bias. Since most data for flood analysis (e.g., elevation data, rainfall distribution fields, remote sensing image) come in this format, CNNs have been increasingly employed by the research community in the recent years. While most papers consider standard CNNs, there are a few which employ 1D-CNNs (e.g., Dong et al., 2021; Guo et al., 2021; Liu et al., 2021) and 3D-CNNs (e.g., Wang et al., 2020b; Fang et al., 2020a). 1D-CNNs consider as input a hyetograph or a hydrograph of a certain event, while 3D-CNNs

consider raster files stacked upon each other. Regarding the architectures, different papers for flood inundation consider an encoder-decoder structure for image segmentation and classification (e.g., Nemni et al., 2020; Hashemi-Beni and Gebrehiwot, 2021; Liu et al., 2019). For such papers, the input is a satellite image of a flood and the output is its classification in flooded and non-flooded areas. This architecture allows the models to increase its performance since it can retain high frequency details in the segmented images (Badrinarayanan et al., 2015).





Guo et al. (2021) and Löwe et al. (2021) use a convolutional encoder-decoder structure for flood hazard mapping to embed a
rainfall hyetograph in the latent space. In this way, they can consider both spatial and temporal data within the same framework.

     RNNs have been mostly employed to model temporally-varying floods, where they can exploit best their sequential inductive
bias. However, they remain the least common choice of DL architecture for spatial flood analysis. Most papers apply RNNs
on a time series, such as a hyetograph or a hydrograph (e.g., Kao et al., 2021; Zhou et al., 2021). Some papers, instead,
consider spatial sequentiality by reshaping the original raster data into vectors (e.g., Fang et al., 2020a; Panahi et al., 2021;
Lei et al., 2021). For example, Fang et al. (2020a) extract, for each pixel, its neighboring pixels in a $3 \times 3$ window and then
convert them into a vector based on spatial contiguity. However, this operation introduces arbitrariety in the sequential order
chosen for arranging the input pixels, since it is independent of the underlying topography. In fact, Panahi et al. (2021) and Lei
et al. (2021) show that these models underperform when compared with CNNs. Among the different RNN layers, most works
consider LSTM units (Kao et al., 2021; Zhou et al., 2021; Fang et al., 2020a) but simple recurrent units (Panahi et al., 2021;
Huang et al., 2021a) and GRUs (Dong et al., 2021) have also been employed. Some papers analyzed the potential of RNNs
in combination with other techniques. Kao et al. (2021) use an encoder-decoder architecture to forecast flood features based
on rainfall patterns. The encoder and the decoder steps are composed of fully-connected layers, while a LSTM is present in
the latent space to process rainfall data. Zhou et al. (2021) identify representative spatial locations in the study area. Then, an
LSTM is trained to simulate the water levels' evolution in time at each location. A water surface is ultimately determined by
interpolating the water depth at those points. Dong et al. (2021) combine 1D-CNNs and RNNs on an urban channel network.
The model takes as input the channels' properties, such as their cross-sections, and rainfall and water level measures, taken
from sensors in the network. This input is then given in parallel to a 1D-CNN and to a GRU whose output is then combined to
predict the temporal evolution of the flood. Hu et al. (2019) deploy the LSTM model in a lower-dimensional space, obtained
via proper orthogonal decomposition and singular value decomposition. The model then requires fewer data to be trained.

### 3.2.5   Performance assessment

This section discusses different approaches for assessing the performance of the DL models, i.e., how well they match the out-
comes of traditional and machine learning models. Flood susceptibility and inundation models are compared with techniques
such as frequency ratio (Popa et al., 2019), a type of MCDA model; the soil conservation service runoff model (Jahangir et al.,
2019), a hydrologic model; and automatic threshold model (Nemni et al., 2020), a histogram-based model. They are also com-
pared with machine learning techniques, such as support vector machines (e.g., Sarker et al., 2019; Gebrehiwot et al., 2019;
Zhao et al., 2020b), random forest (e.g., Darabi et al., 2021; Zhao et al., 2020b), adaptive neuro-fuzzy inference system (Panahi
et al., 2021), deep boost (e.g., Chakrabortty et al., 2021a; Ahmed et al., 2021), and radial basis function (Nogueira et al., 2017).
DL models show to outperform both traditional and ML models in terms of the accuracy of the results. Flood hazard models,
instead, are compared against numerical models, since they act as surrogate models. Thus, their main purpose is to increase
computational speed while maintaining low prediction errors.

     There are also a few papers that compared different DL models. Huang et al. (2021b) compared MLPs with RNNs, while
Fang et al. (2020a) showed that MLPs were outperformed by the more inductive-biased approaches such as RNNs, 1D-CNNs,



and 3D-CNNs. Wieland and Martinis (2019) showed that CNNs widely outperform MLPs, as expected, because of their
inductive biases capabilities. Besides accuracy, the number of parameters and the data requirements are important factors
when comparing DL models. A higher number of parameters results in better performances but may also lead to overfitting, a
condition where the model decreases its performance on the testing data. Hence, when deployed in similar settings such model
would perform drastically worse. Moreover, data is not always available leading to possibly unfair comparisons between models
with different data budgets. As such, the same model may give different outcomes depending on the considered case.

In supervised learning, we distinguish between regression and classification problems, depending on whether the target
values to predict are continuous (e.g., water depth) or discrete (e.g., flooded vs non-flooded area), respectively. Depending on
the task, we employ a different set of metrics to evaluate model performances.

Regression metrics are a function of the differences, or residuals, between target and predicted values. The most common
metrics include the root mean squared error (RMSE), the coefficient of determination ($R^2$), and the mean average error (MAE).
RMSE and MAE improve as they approach zero, while $R^2$ improves as it approaches one. In general, RMSE is preferred to
MAE since it minimizes the standard deviation of the errors, thus decreasing the presence of extreme outliers. However, since
these metrics are averaged on a domain, their comparison across different works requires careful attention.

Classification tasks can be either binary (e.g., predict flooded and non-flooded locations) or multi-categorical (e.g., clas-
sifying between permanent water bodies, buildings, and vegetated areas), according to the output number of classes. In the
following discussion, we focus on the former, with concepts extending to the second case. When computing binary classifi-
cation metrics, flooded areas are generally represented as positive class, while non-flooded areas as negative class. The most
common metrics for flood modelling are accuracy, recall, and precision, followed by other indices such as the area under the
receiver operator characteristic curve. Accuracy represents the number of correct predictions over the total. While popular
and easy to implement, this metric is inappropriate for imbalanced datasets, where some categories are more represented than
others. For example, if test samples feature an average 90% non-flooded area, a naïve model constantly predicting no flooding
would reach 90% accuracy, despite having wrong assumptions. Furthermore, since it may be better to overestimate a flooded
area than to underestimate it, one could resort to metrics such as recall that account for false negatives and thus penalize models
that cannot recognize a flooded area correctly. However, when used alone, recall can lead to similar issues to those described
for accuracy, e.g., yielding a perfect score for a model always predicting the entire domain as flooded. Thus, for an exhaustive
understanding of the model's performance, one should also consider metrics accounting for false positives, i.e., where the
model misclassifies non-flooded areas as flooded. There are several possible metrics, such as the F1 score, the Kappa score, or
the Matthews correlation coefficient, each with their drawbacks and benefits (e.g., Wardhani et al., 2019; Delgado and Tibau,
2019; Chicco and Jurman, 2020). A reasonable choice is the F1 score, which is the geometric mean of recall and precision,
and it thus equally considers both false negatives and false positives. Another good example is the ROC (Receiver Operating
Characteristic) curve that describes how much a model can differentiate between positive and negative classes for different
discrimination thresholds (Bradley, 1997). The Area under the ROC curve (AUC) is often used to synthesise the ROC as a
single value. However, the AUC loses information on which parts of the dataset the model performs best. For this reason, one





should always interpret these results carefully, especially when comparing different studies. Our purpose here is to show that, for the same case study, DL tends to outperform traditional models.

For surrogate models, the comparison is also performed in terms of their speed-up, which is determined as the ratio between the simulation time of the numerical model and the simulation time of the DL model. For a correct comparison, the training time of the DL model must be considered as well in this analysis. However, this was done only by a few papers (e.g., Guo et al., 2021; Kabir et al., 2020; Jacquier et al., 2021).

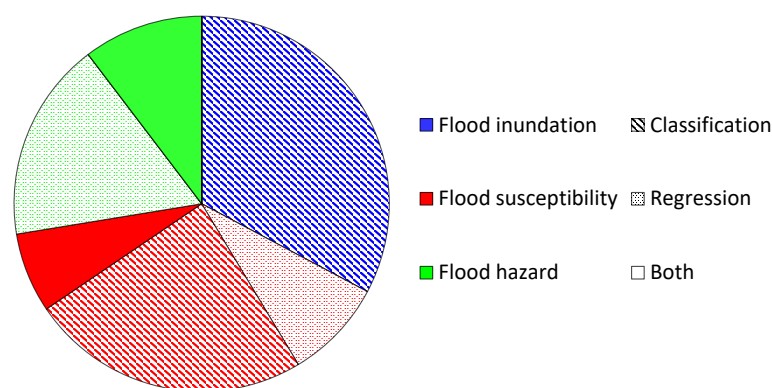

**Figure 7.** Distribution of the comparison metrics per type of application. The colours represent the different types of applications, while the patterns represent the considered metrics.

## 3.3    Deep Learning for flood inundation

Flood inundation maps determine the extent of a flood, during or after it has occurred. The objective is to determine flooded and non-flooded areas from a picture of a flood. This classification is usually binary (e.g., Peng et al., 2019; Nemni et al., 2020) but it can also be extended to include permanent water bodies (e.g., Sarker et al., 2019) (see the example Fig. 1a), vegetation (e.g., Ichim and Popescu, 2020), buildings (e.g., Hashemi-Beni and Gebrehiwot, 2021), and more (e.g., Muñoz et al., 2021). All the types of floods were well represented for this application but flash floods (Fig. 5). We attribute this to the limited frequency of
observation of most remote sensing techniques.

Regarding the spatial scale, most papers focused on local and regional scales. Availability of remote sensing at wider scales is increasingly higher (e.g., Observatory); however, this seems to be only partially considered. A plausible reason is the limited frequency of observation of the satellites. High temporal remote sensing imagery has a low spatial resolution. Few papers tackle this issue by increasing the resolution of the predicted flood maps, via a neural network, with a technique known as *super-*
*resolution* (e.g., Li et al., 2015, 2016b). Super-resolution enhances the quality of an input low-resolution image (Yang et al.,





2019a). These papers show that MLPs improve the accuracy of super-resolution mapping, with respect to other techniques such as spatial attraction models. We argue that further improvements of super-resolution could be obtained by employing CNNs, which lend themselves naturally for such tasks, as demonstrated by applications in similar fields (Ma et al., 2019).

### 3.3.1 DL architecture

As the task of recognizing floods from a picture can be regarded as an image segmentation task, the most used deep learning models are based on CNNs. There are also a few earlier papers that use MLPs (e.g., Li et al., 2016a; Amini, 2010) because CNNs were not yet adopted by researchers of the field. Dong et al. (2021) use a combination of RNNs and 1D-CNNs to determine the temporal evolution of flooded and non-flooded nodes in an urban channel network, as described previously. In this case, the choice of recurrent and 1D-convolutional layers is well motivated due to their temporal inductive bias.

### 3.3.2 Input and output data

Satellite data is the most used input for flood inundation applications (e.g., Sarker et al., 2019; Peng et al., 2019; Nogueira et al., 2017). Other input data sources include unmanned aerial vehicles data (UAV) (e.g., Gebrehiwot et al., 2019; Ichim and Popescu, 2020), hydrographs (e.g., Hou et al., 2021) and DEMs (e.g., Hashemi-Beni and Gebrehiwot, 2021; Muñoz et al., 2021). Only Dong et al. (2021) differ from the other papers by considering sensors in place of flood pictures. Inundation maps

produced by 3D numerical models are also used as target prediction (Muñoz et al., 2021). The results from the numerical model can be used as a detailed reference for the DL model. Satellite data and UAV imagery are both remote sensing data that represent a flood event seen from above. The main differences concern the scale, the resolution, and the availability. UAVs are applicable only for small areas but their resolution is better than satellite data. UAVs can be readily used but may be unavailable in certain areas. On the other hand, satellite data is available worldwide but its frequency of observation can be limiting. When

combining information from different sources, the input data have different resolutions, leading to possible problems for some deep learning models, which take fixed-size inputs. One way to integrate different data resolutions is by data fusion (e.g., Muñoz et al., 2021). This process allows the creation of more consistent, accurate, and useful information than that provided by any individual data source.

### 3.3.3 Performance assessment

Since flood inundation mapping determines which cells of the flood picture are represented as flooded or not, the task is regarded as a classification problem, as confirmed by the metrics used (Fig. 7). The selected papers often use several metrics (see Table A1) but for clarity we consider a single metric for each work. The metric selection depends on the employed ones and follows the considerations presented in Section 3.2.5, with preference for metrics such as F1, AUC, or recall, if available. Deep learning models have consistently shown improved performances in terms of the selected metrics (Table 3). Li et al. (2015)

and Li et al. (2016b) compare optimization techniques with and without MLPs for super-resolution-based flooding. They show that a DL model slightly increases the performances. This may be because the models are based on MLPs and thus neglect any





spatial structure in the data, which could be considered, instead, by CNNs. Most CNN models show noticeable improvements with respect to traditional threshold methods, such as the Normalized Difference Water Index (NDWI) and automatic threshold model (ATM) (e.g., Wieland and Martinis, 2019; Isikdogan et al., 2017; Nemni et al., 2020), and with respect to machine

learning models such as random forest (RF) and support vector machine (SVM). This reflects similar results obtained in image detection tasks (Badrinarayanan et al., 2015).

**Table 3.** Performance of the deep learning and comparison with reference models for flood inundation.

| Case study size [$km^2$] | Deep learning | | Comparison | | Reference |
|---|---|---|---|---|---|
| | Model | Metric | Model | Metric | |
| 2.2 | MLP | Accuracy = 70% | ML | Accuracy = 57% | Amini (2010) |
| 225 | MLP | Accuracy = 86.1% | SAM | Accuracy = 83.9% | Li et al. (2016b) |
| 3600 | MLP+PSO | Accuracy = 81.6% | PSO | Accuracy = 79.7% | Li et al. (2016a) |
| 5625 | MLP+GA | Accuracy = 81% | GA | Accuracy = 79.3% | Li et al. (2015) |
| 0.00016 | CNN | Precision = 0.927 | - | - | Hou et al. (2021) |
| 0.01 | CNN | Accuracy = 95.5% | SVM | Accuracy = 87.4% | Gebrehiwot et al. (2019) |
| 0.9 | CNN | IoU = 84% | SVM+RBF | IoU = 83% | Nogueira et al. (2017) |
| 5.2 | CNN | Accuracy = 92.4% | RG | Accuracy = 91.8% | Hashemi-Beni and Gebrehiwot (2021) |
| 10 | CNN | Accuracy = 96.4% | RF | Accuracy = 87.3% | Ichim and Popescu (2020) |
| 59.3 | CNN | F1 = 0.955 | RF | F1 = 0.922 | Peng et al. (2019) |
| 59.3 | CNN | F1 = 0.90 | RF | F1 = 0.84 | Wieland and Martinis (2019) |
| 200 | CNN | F1 = 0.975 | - | - | Muñoz et al. (2021) |
| 237 | CNN | F1 = 0.90 | NDWI | F1 = 0.70 | Isikdogan et al. (2017) |
| 10895 | CNN | F1 = 0.94 | M1 | F1 = 0.78 | Kang et al. (2018) |
| 23300 | CNN | F1 = 0.88 | - | - | Liu et al. (2019) |
| 25000 | CNN | F1 = 0.92 | ATM | F1 = 0.71 | Nemni et al. (2020) |
| 31450 | CNN | Recall = 62.8% | SVM | Recall = 25% | Sarker et al. (2019) |
| 4600 | RNN+CNN | F1 = 0.77 | CNN | F1 = 0.734 | Dong et al. (2021) |

ML = maximum likelihood; SAM = spatial attraction mode; PSO = particle swarm optimization; GA = genetic algorithm; SVM = support vector machine; RBF = radial basis function; RG = region growing; RF = random forest; NDWI = normalized difference water index; ATM = automatic threshold model

## 3.4 Deep Learning for flood susceptibility

Flood susceptibility determines the tendency to flooding of a study area based on its physical characteristics and given a set of known past flood events. This is done by assigning to each location a level of susceptibility ranked from low to high (see

Fig. 1b). The susceptibility depends on the distribution of the inputs, often called flood conditioning factors, in function of





recorded past flood events. The deep learning model then computes, for each point in the area, a score from 0 (non-flooded) to 1 (flooded). These scores are finally divided into several classes, generally using the natural (Jenks) breaks method (e.g., Fang et al., 2020a; Wang et al., 2020b; Khoirunisa et al., 2021), to obtain a susceptibility map. An exception is given by Jahangir et al. (2019) and Kia et al. (2012), which train their models to predict discharge values and then use a GIS model for the

mapping. In both cases, the model performs well when the recorded flood events occur in the predicted high susceptibility areas.

There exist DL-related applications for all types of flood (see Fig. 5). Furthermore, Fig. 6a shows that most of the works are concerned with regional or wider scales (e.g., Tien et al., 2020; Panahi et al., 2021; Khosravi et al., 2020). This is expected since susceptibility mapping gives a qualitative estimate of which locations are prone to flooding. Operating on small scales

may thus be limiting, both in terms of data availability and applicability for prevention strategies. The data requirements for an accurate estimate would probably be too high for a small area.

### 3.4.1 DL architectures

Most papers use MLP and CNN. Models based on MLPs consider single points or pixels as inputs (Tien et al., 2020; Ahmadlou et al., 2021; Khoirunisa et al., 2021), while CNNs consider the whole raster files (Zhao et al., 2020b; Khosravi et al., 2020;

Wang et al., 2020b). Since MLPs lack inductive bias they provide less coherent results, meaning that the variation among neighboring cells can be high. This is partially solved by coupling the MLP architecture with other statistical techniques, such as frequency ratio (e.g., Darabi et al., 2021; Popa et al., 2019; Costache et al., 2020). Instead, CNNs have a spatial inductive bias, thus they inherently consider the structure of the input, providing more coherent flood maps (e.g., Khosravi et al., 2020). However, Wang et al. (2020b) and Liu et al. (2021) show that 1D-CNNs, which perform convolution on the input features for

each domain's cell, are not suited for this problem, as they do not properly leverage any inductive bias. Some works showed that deep belief networks (DBN), an unsupervised variation of MLPs, could outperform standard MLPs in flood susceptibility mapping (e.g., Shirzadi et al., 2020; Pham et al., 2021).

### 3.4.2 Input and output data

The inputs for the deep learning models are several. We distinguish between five input typologies:

1. *topographical* inputs, which are derived from a digital elevation model, such as elevation, slope, and aspect;

2. *meteorological* inputs, related to the hydrological characteristics and derived from measuring stations and satellites, such as rainfall distribution and frequency;

3. *geological* inputs, related to the properties of the soil, such as lithology and soil type;

4. *geographical* inputs, related to observable surface characteristics and obtained through remote sensing, such as land use

and normalized difference vegetation index;

5. *anthropogenic* inputs, related to the presence of human-made environments, such as distance from roads.





Topographical data were the most frequent type of input. Most papers also performed a statistical analysis to determine which factors influenced the most the final results: on average the most important factors were slope, land use, aspect, terrain curvature, and distance from the rivers (e.g., Khosravi et al., 2020; Fang et al., 2020a; Popa et al., 2019; Costache et al., 2020). A complete

list of inputs is reported in the Appendix (Fig. B1).

As output data, most papers considered a flood inventory map, given by a set of flooded and non-flooded locations. The flooded locations were derived from measurements and records taken from remote sensing and stations, while non-flooded locations were taken randomly from locations with no previous flood record.

### 3.4.3    Performance assessment

In flood susceptibility analysis, both classification and regression metrics are adopted (Fig. 7). While classification metrics are used to identify flooded or non-flooded areas, the purpose of regression metrics is often omitted unless the reference target is a discharge hydrograph (Jahangir et al., 2019; Kia et al., 2012). Both types of metrics are used in few papers (e.g., Panahi et al., 2021; Khosravi et al., 2020). Because of the problem's setup, classification metrics are more reliable in the performance assessment. Following the considerations in Section 3.2.5, we selected as preferable metric AUC, also because of its frequent

availability for flood susceptibility mapping. For all the papers with comparisons, deep learning models consistently showed improved performances with respect to the reference models with few exceptions (Table 4). Deep boost (DB) is a machine learning algorithm based on deep decision trees (Cortes et al., 2014) which could slightly outperform MLP in few works (Ahmed et al., 2021; Chakrabortty et al., 2021b). Combining optimization algorithms, such as particle swarm optimization, with MLPs, to improve the training, has a limited effect on the performance improvement (Kalantar et al., 2021; Ngo et al.,

2018). Moreover, CNNs increase the performance with respect to traditional models more than MLPs. Fang et al. (2020a) show that encoding spatial sequentiality with LSTMs work slightly better than 1D-CNNs and 3D-CNNs, however they avoid the comparison with 2D-CNNs.

### 3.5    Deep Learning for flood hazard

Flood hazard predicts the depth, velocity, and extent of floods. This application produces maps which evaluate for a certain

event its maximum inundation (e.g., Guo et al., 2021; Berkhahn et al., 2019; Löwe et al., 2021) or how it evolves in time (e.g., Lin et al., 2020a; Zhou et al., 2021). While most studies consider the probability of different events, using return periods (e.g., Kabir et al., 2020; Guo et al., 2021), there are few papers which determine the water depth map for a single event (e.g., Hu et al., 2019; Chang et al., 2010). However, no papers were identified to predict the flow velocities. Since the simulation results are taken as ground-truth data for training, deep learning models for flood hazard mapping are used as surrogate models in

place of numerical models.

The most studied types of floods are river and urban floods. As regards the spatial scale, the models are carried out at local and regional scales. This is probably due to the computational burden of performing several simulations at larger scales to train the deep learning model.





**Table 4.** Performance of the deep learning and comparison with reference models for flood susceptibility.

| Case study size [$km^2$] | Deep learning | | Comparison | | Reference |
|---|---|---|---|---|---|
| | Model | Metric | Model | Metric | |
| 27 | MLP+ensemble | AUC = 0.847 | RF | AUC = 0.821 | Darabi et al. (2021) |
| 132 | MLP | $R^2 = 0.802$ | - | - | Khoirunisa et al. (2021) |
| 147 | MLP+PSO | AUC = 0.98 | MLP | AUC = 0.96 | Kalantar et al. (2021) |
| 207 | MLP | $R^2 = 0.82$ | SCS | $R^2 = 0.71$ | Jahangir et al. (2019) |
| 465 | MLP | AUC = 0.917 | PSO | AUC = 0.929 | Chakrabortty et al. (2021b) |
| 1128 | MLP | AUC = 0.901 | DB | AUC = 0.917 | Ahmed et al. (2021) |
| 1465 | MLP | AUC = 0.960 | SVM | AUC = 0.936 | Tien et al. (2020) |
| 1510 | MLP | AUC = 0.970 | SVM | AUC = 0.960 | Ngo et al. (2018) |
| 2600 | MLP+AHP | AUC = 0.953 | MLP+FR | AUC = 0.942 | Costache et al. (2020) |
| 4673 | MLP | AUC = 0.93 | DB | AUC = 0.96 | Chakrabortty et al. (2021b) |
| 5264 | MLP+FR | AUC = 0.97 | FR | AUC = 0.937 | Popa et al. (2019) |
| 12050 | MLP | AUC = 0.974 | - | - | Ahmadlou et al. (2021) |
| 132000 | MLP | $R^2 = 0.98$ | - | - | Kourgialas and Karatzas (2017) |
| 131 | CNN | AUC = 0.90 | RF | AUC = 0.78 | Zhao et al. (2020b) |
| 605 | CNN | AUC = 0.84 | RNN | AUC = 0.82 | Lei et al. (2021) |
| 1543 | CNN | AUC = 0.937 | SVM | AUC = 0.883 | Wang et al. (2020b) |
| 12000 | CNN | AUC = 0.832 | ANFIS | AUC = 0.70 | Panahi et al. (2021) |
| 1649195 | CNN | AUC = 0.75 | - | - | Khosravi et al. (2020) |
| 90016 | CNN+FMV | AUC = 0.912 | SVM-FMV | AUC = 0.898 | Liu et al. (2021) |
| 1543 | LSTM | AUC = 0.965 | 3D-CNN | AUC = 0.956 | Fang et al. (2020a) |

PSO = particle swarm optimization; SCS = soil conservation system model; SVM = support vector machine;
AHP = analytic hierarchy process; RF = random forest; FR = frequency ratio; DB = deep boost; ANFIS = adaptive
neuro-fuzzy inference system; FMV = fuzzy membership value

### 3.5.1 DL architecture

The deep learning models are mainly based on MLPs and RNNs. In particular, RNNs are applied when a spatial-temporal estimation of the water depths is performed. CNNs were initially discarded but are used more in recent years (e.g., Guo et al., 2021; Löwe et al., 2021; Kabir et al., 2020). Hu et al. (2019) and Jacquier et al. (2021) use a LSTM and a MLP, respectively, in combination with a reduced order modelling framework. In the first case, the DL model is applied on the reduced space, while in the latter DL is used as surrogate for the decomposition method.





### 3.5.2 Input and output data

The inputs are hyetographs, which represent the rainfall precipitation or intensity in time (e.g., Berkhahn et al., 2019; Kao et al., 2021; Guo et al., 2021) or hydrographs, which represent the discharge in time (e.g., Chu et al., 2020; Zhou et al., 2021; Lin et al., 2020a). Other inputs such as the DEM and the roughness coefficient, also used for numerical models, are sometimes considered as additional inputs (e.g., Guo et al., 2021; Chang et al., 2010; Huang et al., 2021b). Löwe et al. (2021) performed a forward selection to identify relevant topographic variables, showing that aspect and local depressions improve the model's prediction for urban floods.

The output is a water depth map. For the datasets, it is obtained via numerical models based on the 2D shallow water equations. 1D, 1D-2D, and 3D models are also used (Kao et al., 2021; Chang et al., 2010; Hu et al., 2019). The main reason why numerical models are used is to simulate events that have never occurred or have never been observed, such as floods with high return periods. Even though observed data were not employed, they could be used in future research to corroborate the transferability of such methods. When training only on the numerical models' predictions, the deep learning models' results are limited in accuracy by the numerical models' one, i.e., if the numerical model does not represent reality so will the DL model. Thus, when the model is deployed on real data, there may also be some generalization issues caused by the difference between the training and testing data. The inclusion of real measured data may thus also improve the accuracy with respect to numerical models.

### 3.5.3 Performance assessment

In flood hazard, regression metrics are used to evaluate the water depth, while classification metrics are used to evaluate the flood extent, as done for flood inundation (Fig. 7). While for flood susceptibility and inundation DL models were used to improve the performances, in flood hazard their main focus is to improve the speed, while still maintaining reasonably low errors with respect to the numerical predictions. This is highlighted in Table 5, for all papers which provide information on computational times of both numerical and deep learning models. However, the comparison of speed-up across different papers is often unrealistic since it depends on the number of performed numerical simulations and on the type of numerical model. A similar consideration persists for the error scores, as they depend on the scale of the case study and on its resolution. A final remark regards the loss function employed in the training of the DL models. The minimization of the squared errors does not guarantee that the solution will have physical meaning. For flood hazard mapping a possible solution is then to enforce the conservation of the mass or momentum equations by adding such terms in the loss function. This provides additional biases on the predicted solution and was shown to increase its performance in representing the numerical models (e.g., Zhang et al., 2021).





**Table 5.** Performance of the deep learning and comparison with numerical models for flood hazard.

| Case study size [$km^2$] | Deep learning Model | Numerical Model | Comparison measure | Speed-up | Reference |
|---|---|---|---|---|---|
| 0.6 | MLP | 2D | RMSE = 0.0013 m | 1000x | Berkhahn et al. (2019) |
| 7.7 | MLP | 2D | RMSE = 0.16 m | 100x | Chu et al. (2020) |
| 21 | MLP+clustering | 2D | RMSE < 0.3 m for 99.7% of domain | - | Huang et al. (2021a) |
| 31 | MLP | 1D-2D | MAE = 0.06 m | 1000x | Chang et al. (2010) |
| 92 | MLP | 2D | RMSE < 0.3 m for 82% of domain | - | Lin et al. (2020a) |
| 92 | MLP | 2D | MSE < 0.2 $m^2$ for 91% of domain | - | Lin et al. (2020b) |
| - | MLP+ROM | 2D | RE = 2.8% | 33x | Jacquier et al. (2021) |
| 10 | CNN | 2D | MAE = 0.0012 m | - | Hosseiny (2021) |
| 10.5 | CNN | 2D | MAE < 1 m for 93% of domain | 2000x | Guo et al. (2021) |
| 14.5 | CNN | 2D | RMSE = 0.11 m | 38x | Kabir et al. (2020) |
| 72 | CNN | 2D | RMSE = 0.22 m | - | Yokoya et al. (2020) |
| 400 | CNN | 2D | RMSE = 0.08 m | - | Löwe et al. (2021) |
| 18.5 | LSTM+ROM | 3D | RMSE = 0.01 m | 1500x | Hu et al. (2019) |
| 271 | LSTM | 2D | RMSE = 0.08 m | - | Kao et al. (2021) |
| 1479 | RNN | 2D | RMSE = 0.056 m | 21x | Zhou et al. (2021) |

ROM = reduced order modelling

## 4 Knowledge gaps

We identified knowledge gaps regarding the applications in flood management, usability, generalization, modeling limitations, and data availability. Some other minor gaps were shown in the previous section. Based on these gaps, future research directions are proposed in Section 5.

### 4.1 Flood applications and usability

Deep learning has proven useful for assessing flood-prone areas from the location of past events, identifying flooded areas
from remote sensing images, and working as a surrogate model for numerical simulations. Nonetheless, each of the presented maps has its own limitations. Susceptibility maps provide only qualitative results and rely on recorded flood events. Therefore, limited recorded data may lead to incorrect predictions. Moreover, it is important to design an appropriate model to integrate heterogeneous environmental information. Inundation maps mostly consider real events, thus they suffer from the acquisition





method's problems. For example, satellites struggle to extract information below clouded areas (e.g., Meraner et al., 2020).
Hazard maps, instead, are limited by the accuracy of the underlying numerical simulator. This leads to explore also other applications within this field that could benefit from deep learning models. In particular, we address two flood management applications, *flood risk* and *real-time flood warning*. We also define two desired types of maps, *flood arrival time maps* and *probabilistic hazard maps*. Then, we discuss *dam and dike breach flood* events.

   Flood risk combines the probability that a certain event occurs with the associated consequences, such as economic impacts
or loss of life. The expected annual loss is a common measure obtained from flood risk assessment and depends on (i) flood hazard, given by event-specific flood characteristics, such as water depth and flow velocity, (ii) exposure, related to the elements at risk, such as buildings and critical infrastructures, and (iii) vulnerability, i.e., the inability of a system to withstand the effects of the event, given, for example, by intensity-damage curves. Flood risk maps are obtained by combining flood hazard maps with damage models. Other approaches are based on MCDA, since the exact flood magnitude and damage are
uncertain (de Brito and Evers, 2016). This is done by incorporating various factors that determine flood risk, such as hazard, the performance of defenses, topography, and exposure. However, MCDA is based on expert knowledge and is thus subjective. DL models solve this issue and can also yield a higher accuracy, as shown for flood susceptibility mapping. Thus, DL-based approaches could provide alternative methods for assessing flood risk. In addition to the inputs used for flood susceptibility, such as elevation and land use, flood risk mapping may require also other inputs such as population density, spatial estimates of
economic value, and building types. Up to now, only Chen et al. (2021) combined DL and flood risk assessment. They showed that ML and DL approaches can estimate flood risk at regional scale, but do not compare their results against other methods, such as MCDA. One drawback of their approach is that the resulting maps were qualitative, while quantitative results should be preferable for risk assessment.

   Real-time flood warning is another application that has not been widely addressed. This is needed by local authorities to
inform when and where a flood may occur. While several papers mention real-time prediction, most can be used only after the event has occurred, since they require as input the complete hyetograph or hydrograph of the event. There are a few examples based on RNNs which could forecast floods in near real-time using sensors (Kao et al., 2021) and rainfall distribution (Dong et al., 2021). However, few situations are covered and, thus, more research should focus on filling this gap. An alternative method is to predict the rainfall in real-time and then retrieve the corresponding water depth map by using a similarity measure
on a large dataset of previous simulations (Chang et al., 2020). However, such a solution may be challenging because of the large storage requirements. Using DL for surrogate modeling instead showed substantial speed improvements, thus allowing for real-time simulations and forecasts. Similar achievements have already been obtained for rainfall now-casting, where the deep learning models can accurately forecast the near-future rainfalls (e.g., Shi et al., 2015; Ravuri et al., 2021).

   Arrival time maps estimate the time employed by a flood to reach a certain water depth threshold. They can encode both
spatial and temporal information in the same map. So, for a practitioner, they carry at one place detailed information not only on where to intervene but also when to execute mitigation measures. Despite these promises, they have been seldom used in flood management; consequently, they have also not been exploited with DL methods. Using DL for arrival map estimation may be a promising direction to identify critical infrastructure and set up corresponding evacuation plans in real-time. This is





because DL has shown potential for surrogate modeling (see Table 5) and because arrival maps can be obtained from flood hazard maps taken over different time intervals of a flood event. This application may be particularly important for exceptional flood events, such as dike breaches and dam breaks, where little forecast can be made until a failure initiates (Yakti et al., 2018).

Probabilistic hazard mapping captures the model uncertainty related to its inputs and outputs. As pointed out by Di Baldassarre et al. (2010), uncertainties can result in deterministic maps which are only spuriously accurate. But probabilistic maps can account for the uncertainties by assigning a probability of flooding to each domain element. This analysis is generally carried out with probabilistic methods such as Monte Carlo simulations (e.g., Papaioannou et al., 2017). However, since they require a vast amount of simulations, only simpler numerical models are used. DL models could be used as surrogates to speed up computation and improve the accuracy of the simpler models. Nonetheless, brute-force simulations, such as Monte Carlo, may require up to hundreds of thousands of simulations to obtain a satisfactory measure of the uncertainty (Liu, 2017). Thus, we need models that can intrinsically work with probabilistic input distributions of parameters.

Dam break and dike breach floods concern a relevant category of flood events that has been poorly approached with deep learning models. The motivation is probably related to the rarity of such events and the complexity of the phenomena. However, their catastrophic and unexpected effects make their modeling necessary in several situations. Moreover, the effect of flood defenses' failure is often disregarded, also because location and modality of possible failures are uncertain. A common way to include the failure of structures is to investigate all possible combinations of locations and boundary conditions, but it can be constrictive both for time and storage capacities. Probabilistic hazard mapping may be a relevant application to include the uncertainty in the failure probability of the flood defense (Domeneghetti et al., 2013).

## 4.2 Generalization

Generalization refers to the capacity of a model to extrapolate from a training dataset into unseen testing data. This means that a DL model can correctly predict scenarios unused in its development. This property is particularly relevant because training requires data, model set-up, and time. In the context of flood modelling, there are two main generalization objectives: (i) boundary conditions, i.e., different rainfall events, and (ii) topographical changes, i.e., different case studies. However, the transference between different areas is challenging for DL models because of the difference in input and output data. In fact, except for flood inundation mapping, most reviewed papers focused on generalizing different boundary conditions (e.g., Guo et al., 2021; Berkhahn et al., 2019). Instead, only a few papers tested the model on areas not considered during training. Löwe et al. (2021) could generate flood hazard maps for unseen areas within the same study region as the training dataset, as there was little variability of inputs and outputs. Zhao et al. (2021b) instead pre-trained a model for flood susceptibility on an urban area and then used it for another similar area. They showed that pre-training improves predictions with respect to a model trained from scratch, both in cases of low and high data availability. These works show that such approaches are in their infancy and have been tested on limited datasets. A DL model which cannot generalize to new areas has to be trained every time for a new study case. Thus, it may have limited advantages over a hydraulic model, since it requires more effort, data, and time. Instead, a general DL model which can generalize to new areas could emphasize the advantages over numerical models. This





concept was experimented also for rainfall-runoff modeling where DL models outperformed state-of-the-art alternatives in the prediction of ungauged basins in new study areas (Kratzert et al., 2019b).

## 4.3   Modeling limitations

Complex interactions with the natural and built environment, such as dikes or buildings, are difficult to include in deep learning models. Kabir et al. (2020) showed that flood defenses can be included if they are present in the simulations used for training and testing. However, no solution presented so far can directly include new flood defenses in it. Building can be statically included as well in the DEM (e.g., Löwe et al., 2021), but bridges and other hydraulic structures that influence the behavior of the floods may be harder to include, due to their strong influence on the flow path.

## 4.4   Data availability

Deep learning models usually require large quantities of data to achieve good performances. While simulations can provide potentially limitless data, observed data are scarce and depend on the study area. Simulations may also encounter instability issues depending on the numerical schemes and study area. Remote sensing has provided large quantities of data since its vast development in the past decades, but satellite data is still limited by its frequency of observations and dependency on favorable

meteorological conditions. Also, UAVs cannot cover wide areas at once. Precipitation and water depth data are available only in a few locations where the measuring stations are present. Thus, new data sources are needed to overcome these limitations.

Another issue, which emerges also from Section 3.2.5, is the lack of a unified framework to compare different approaches with each other. This can be achieved by creating flood-based benchmark datasets for each mapping application. For flood inundation, some datasets have been already used across different works (e.g., Bonafilia et al., 2020). However, works on both

flood susceptibility and hazard mapping consider different datasets, focusing on different geographic areas or flood types. One possibility could then be to unify different case studies in a single dataset, for each application, allowing to assess the validity of a model more objectively. For flood susceptibility, case studies with the same input availability could be merged in a dataset with many flood types, scales, and geographical areas. A similar reasoning could be made for flood hazard mapping, selecting, for each case study, initial and boundary conditions for specific return periods.

## 5   Future research directions


The present review shows that flood practitioners still need to be up to date with the latest and most successful deep learning models. We suggest that the outstanding identified issues can be approached by resorting to Deep Learning state-of-the-art advancements to our field. As such, we propose future research directions to transfer this knowledge and address the above-identified gaps.





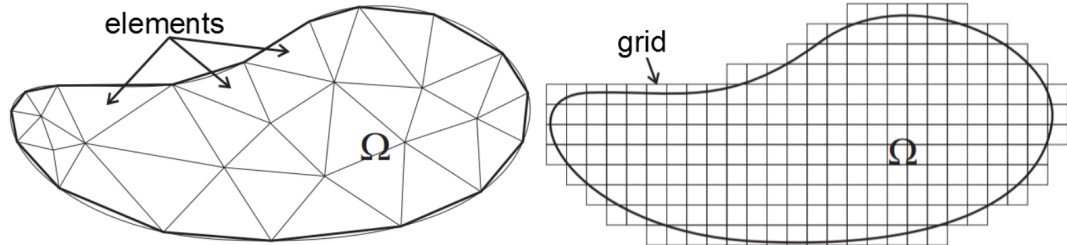

**Figure 8.** The irregular geometrical structure of the mesh allows capturing information in a more efficient way than regular grids by following the properties of the underlying system (picture taken from (Ferreira et al., 2015)).

## 5.1 Mesh-based deep learning

Current deep learning models lack generalization across different case studies, meaning that they can work exclusively for a specific purpose or area. They also cannot represent complex interactions with the natural and built environment. Both issues may depend on the regular grids used in the reviewed papers, which are unable to follow the geometric properties of irregular inputs, as illustrated in Fig. 8. Hence, the model cannot exploit many data patterns, ultimately limiting its generalizability and, for the same motivation, being unable to account for the irregular geometrical structures. Unstructured meshes may solve this problem by discretizing the domain more flexibly (Mavriplis, 1997). A mesh is a structure composed of a collection of nodes, edges, and faces used to discretize a continuous domain. Meshes are commonly used for numerical simulations in many physical systems (e.g., Ferraro et al., 2020; Bomers et al., 2019). Their flexible definition allows to increase the resolution where needed and coarsen it otherwise, ultimately decreasing the computational time and improving efficiency (Candy, 2017). Moreover, they are equivalent to regular grids if the mesh is structured. Thus, the following principles could also be transferred to rasters, if needed. Unstructured meshes, nonetheless, inherit similar problems as those typical of numerical models, such as mesh generation and the need of explicitly defining how each node is connected. Standard DL models, such as CNNs, cannot be applied on meshes. There are currently several lines of work which, instead, can use meshes as a learning framework. They are here referred to as *mesh-based neural networks*. Two highly promising mesh-based approaches for flood applications are *geometric deep learning* and *physics-based deep learning*.

### 5.1.1 Geometric deep learning

Geometric deep learning provides a generic framework to work with any type of data by enforcing symmetries with respect to transformations, such as translations and rotations (Bronstein et al., 2017). Symmetries result in inductive biases, which address the curse of dimensionality by decreasing the required training data (e.g., Wang et al., 2020a) and enabling the processing of different data types, such as meshes. From a flooding perspective, symmetries can be understood and motivated by referring to the example in Section 2.2.1. For instance, analogously to translation, the rotation of a domain should result in an equivalent rotation of the predictions. Among the several geometric deep learning models which can work with meshes, *graph neural*





*networks* (GNN) are the most developed ones. Graphs are structures defined by a set of nodes and edges and can be considered as the underlying skeleton of a mesh. GNNs allow to model data on graphs by considering how its elements are connected (Wu et al., 2021; Gama et al., 2020). They take as input the information encoded in the nodes, in the edges, and in the graph structure, and then process it with neural networks, in a similar way as done by CNNs and RNNs with grid elements and sequential data, respectively. For example, nodes can carry information on the elevation of a point or its boundary conditions, while edges may encode the spatial distance between nodes. Several variations of GNNs exist that give more importance to certain parts of the data by weighting information from different neighbours (Wu et al., 2021; Isufi et al., 2021). There already exist promising works which simulate fluid dynamics with mesh-based GNNs, with increased generalization, accuracy, and stability, with respect to CNNs (e.g., Pfaff et al., 2020; Lino et al., 2021). However, GNNs consider only pairwise geometrical properties as connections between nodes, thus neglecting the mesh structure. Recent developments focused on extending the GNN framework to include it. *Mesh convolutional neural networks* adapt GNNs to include a representation of the local geometry, which preserves the angles between edges (De Haan et al., 2020; Zhou et al., 2020). *Simplicial neural networks* (Yang et al., 2021; Ebli et al., 2020) and *cell complex neural networks* (Bodnar et al., 2021; Hajij et al., 2020), instead, generalize GNNs to higher-order structures. They can consider as well information on triangular and polyhedral elements, which can represent, for example, a flooded area or a volume. This inclusion of the mesh properties in such approaches may further enhance the power of GNNs. Even though they are still in their infancy, their potential for learning on meshes could reveal to be useful also for flood modelling in future research.

### 5.1.2 Physics-based deep learning

While promising, the aforementioned approaches ignore any underlying physical laws present in flood modelling and let the model figure them out. But these physical laws provide additional inductive biases, hence we could include them in modeling to enhance the performance. Physics-based neural networks and neural operators are approaches that account for them.

*Physics-informed neural networks* (PINN) employ physical laws to constraint the model solution (Raissi et al., 2019). The idea is to parameterize a partial differential equations (PDE) solution with a neural network, while keeping the same physical formulation. Then, each partial derivative in the equations is determined via automatic differentiation. Many works have shown the capabilities of PINNs to follow the underlying PDEs in fluid dynamics (e.g., Mao et al., 2020; Yang et al., 2019b). This is relevant also in flood modelling where PDEs such as shallow water equations or the Navier-Stokes equations are employed (e.g., Mahesh et al., 2022). However, PINNs can only be trained for a specific boundary condition (e.g., a specific rain event) and can subsequently only simulate that specific event (Kovachki et al., 2021).

Neural operators, instead, can learn mappings between function spaces, i.e., they learn a whole family of equations (Kovachki et al., 2021). In other words, they can approximate any differential operator. Moreover, since neural operators learn a mapping between infinite-dimensional spaces, they are invariant with respect to the chosen discretization. Thus, their solution is transferable to any mesh resolution. While many approaches have been proposed, such as DeepONets (Lu et al., 2019) or multipole graph neural operator (Li et al., 2020), *Fourier neural operators* (FNO) have currently achieved the best results (Li et al., 2021). In general, the idea is to extract features from the input function, process them in the function space, and,



finally, map them to the output function. In FNOs, the function space is given by the Fourier space, which allows to use fast Fourier transforms, providing faster approximations of the integral operator. Results show that FNOs improve the speed of several PDEs by up to three orders of magnitude. Jiang et al. (2021) used FNOs for simulating sea surface height, showing
increased performance with respect to CNNs, and noticeable speed-up compared to the numerical simulator. Consequently, they could also be used in flood management to overcome computational speed limitations while preserving the underlying physics, allowing also for more reliable real-time flood warning. Thanks to the inductive biases given by the physical laws, both physics-based neural networks and neural operators also require less data.

## 5.2 Probabilistic deep learning

Uncertainties in floods are often determined via probabilistic hazard mapping. These maps show the inundation depths and extents together with their confidence intervals and are traditionally obtained with Monte-Carlo simulations (e.g., Domeneghetti et al., 2013). To avoid brute-force simulations and provide uncertainty guarantees, certain deep learning models can consider uncertainty in the model inputs. An example of these models is *deep Gaussian processes* (DGP). DGP are models composed by the stacking of Gaussian processes (GP), in a similar fashion done in neural networks (Damianou and Lawrence, 2013).
A GP is a collection of random variables whose joint distribution is a Gaussian (Rasmussen, 2003). They benefit from the properties of normal distributions and thus their output can be obtained analytically. The advantage of DGPs over GPs is that they can better extract patterns in data thanks to their increased complexity. DGPs can determine the distribution of the output and could, therefore, be used in probabilistic hazard modeling to determine the range of variation of the predicted flood hazard map. No example of DGPs used for flood mapping exists yet. However, GPs have been used for the statistical estimation of the
correlation between flooding and sea-level rise (Vandenberg-Rodes et al., 2016).

Along with those related to the model's input, uncertainties are also present in the model's prediction. To account for this kind of uncertainty we can use *Bayesian neural networks* (BNN). BNNs are models with stochastic components trained using Bayesian inference. They assign prior distributions to the model parameters to provide an estimate of the model's confidence on the final prediction (Blundell et al., 2015). If, for different parameter sampling, the output is unvaried, then the model has
a good confidence on the prediction and vice versa if different parameters give different results. Jacquier et al. (2021) used BNNs to determine the confidence intervals in flood hazard maps, providing a measure of the model's reliability.

## 5.3 Data augmentation

Even though remote sensing and measuring stations provide noticeable amounts of data, several parts of the world still lack enough data to deploy deep learning models. New satellite missions and added sensor networks throughout the world increas-
ingly provide new data sources (e.g., van de Giesen et al., 2014). But here we focus on how DL itself can be one solution for data scarcity.

The flexibility of DL partially overcomes data scarcity by facilitating the use of a wider variety of data sources. For instance, several papers already employ cameras to detect floods and measure the associated water depth (e.g., Vandaele et al., 2021; Jafari et al., 2021; Moy De Vitry et al., 2019). Structural monitoring with cameras can provide reliable data where it was



previously hard to obtain, such as in urban environments. Social media information can also be used to identify flood events and flooded areas, via tweets or posted pictures (e.g., Rossi et al., 2018; Pereira et al., 2020). In this case, the information's validity and reliability must be considered before its use for real applications. Moreover, the heterogeneity of the sources of these data needs to be carefully taken into account when deploying a DL model.

Another approach can be to generate artificial data to supplement scarce data. This can be done using generative adversarial

networks (GAN), which create new data from a given dataset (Goodfellow et al., 2014). GANs are composed of two neural networks, named generator and discriminator, whose purpose is, respectively, to generate new data and to detect if a given data is real or fake. A trained GAN can produce new fake but plausible data, facilitating data augmentation, i.e., providing more training samples. Interesting applications of GANs could overcome some limitations of satellite data (Lütjens et al., 2020, 2021), predict flood maps (Hofmann and Schüttrumpf, 2021) or meteorological forecasts (Ravuri et al., 2021), and

create realistic scenarios of flood disasters for projected climate change variations (Schmidt et al., 2019). GANs could also be used to generate a plausible urban drainage system or topography for cities that do not have any sewers construction plan or in areas where only low-resolution data is available (e.g., Fang et al., 2020b).

However, GANs are difficult to train (Goodfellow, 2016). *Variational autoencoders* (VAE) are another type of generative model, which can overcome this issue. Differently from standard autoencoders, VAEs model the latent space with probability

distributions that aim to ensure good generative properties to the model (Kingma and Welling, 2013). Once the model is trained, new synthetic data can be generated by taking new samples from the latent distributions. Nonetheless, because of the model's definition, the predictions are less precise than GANs. As such, VAEs and GANs offer a trade-off between the reality of the prediction and the availability of training data.

## 6 Conclusions

This paper presented a review of current applications of deep learning models for flood mapping. The search criteria focused on DL applications in flood mapping from 2010 until 2021, leading to a total of 58 reviewed papers. Our main conclusions can be summarized as follows:

- Flood inundation, susceptibility, and hazard mapping were investigated using deep learning models. Flood inundation considers as the main data images of floods, mostly taken via satellite. The main and most accurate deep learning
models were CNNs. In flood susceptibility, deep learning models consider several inputs, the most important being slope, land use, aspect, terrain curvature, and distance from the rivers. The main deep learning model used were MLPs, often in combination with other statistical techniques, but CNNs provided more accurate results. So far, flood hazard maps estimate the water depth in a study area by using deep learning as a surrogate model for numerical simulations. For this application, there are no deep learning model preferences. However, RNNs are preferable for spatio-temporal
simulations. Regardless of the application, results show that deep learning solutions outperform traditional approaches as well as other ML techniques.





- MLPs and CNNs were the most common type of deep learning model considered in flood mapping, while RNNs were less used. To overcome their lack of inductive biases and achieve good accuracy, MLPs are often coupled with other statistical techniques. On the other hand, thanks to their spatial and temporal inductive biases, CNNs and RNNs were found to regularly outperform other models. DL models used for surrogate modeling provide significant speed-up (up to three orders of magnitude) while maintaining sufficient accuracy.

- Most papers dealt with river and urban floods, while only a few works described applications for flash, coastal, and dam break floods. Case studies were mainly addressed at local or regional scales, arguably due to the availability of high-resolution data. Conversely, the community should further investigate the suitability of DL models for flood applications at larger scales. Concerning the development data, we found that models producing susceptibility and inundation maps rely on the availability of real flood observations. Instead, DL-based surrogate models for hazard mapping require target data from numerical simulations.

This review outlined several knowledge gaps, which can be addressed via deep learning to improve the state of the art of flood mapping. To solve these gaps we proposed possible solutions based on recent advances in fundamental machine learning research:

- Flood risk could be addressed in a similar way as for flood susceptibility by using physical and economical characteristics to obtain a risk map. Flood arrival time maps can provide both spatial and temporal information of a flood event and may be obtained similarly as for flood hazard maps.

- Current deep learning models struggle to generalize across different case studies and regions, meaning that a new model has to be created each time. Moreover, they cannot account for the complex interactions with the natural and built environment. A solution to these problems is to use novel DL architectures that include meshes as learning frameworks. Mesh-based neural networks, such as graph neural networks and neural operators, can consider arbitrarily shaped domains and thus provide the required flexibility to generalize across case studies and model the effects of complex interactions.

- Physics-based deep learning provides a reliable framework for flood modelling since it considers the underlying physical equations. Probabilistic hazard mapping can take advantage of deep Gaussian processes or Bayesian neural networks to determine the uncertainties associated with the model and its inputs.

- DL necessitates large quantities of data which are difficult to collect in several areas of the world. New data sources such as camera pictures and videos, or social media information can potentially be used thanks to deep learning models. Moreover, generative models, such as GANs and VAEs, can be employed to produce synthetic data for such data-scarce regions, based on training data collected elsewhere.

We expect deep learning to be a promising tool to improve and speed up flood mapping. Nonetheless, deep learning models are black-box models, meaning that the underlying operations are unknown. Thus, their deployment in real emergencies has





to be taken cautiously. As deep learning for flood mapping is still novel, we advise its use in critical situations to be always

820    validated by traditional models and expert knowledge, until robust and corroborated models are available. The above concern highlights the main challenge DL models for flood management need to face. However, DL models are still in their infancy and carry the large potential to aid researchers for many applications, especially where traditional models cannot provide sufficient accuracy or speed. In particular, deep learning-based flood mapping approaches could provide an added value for regions with limited data or limited resources to invest in setting up time-consuming hydraulic models.

825    **Appendix A:  Comparison metrics**

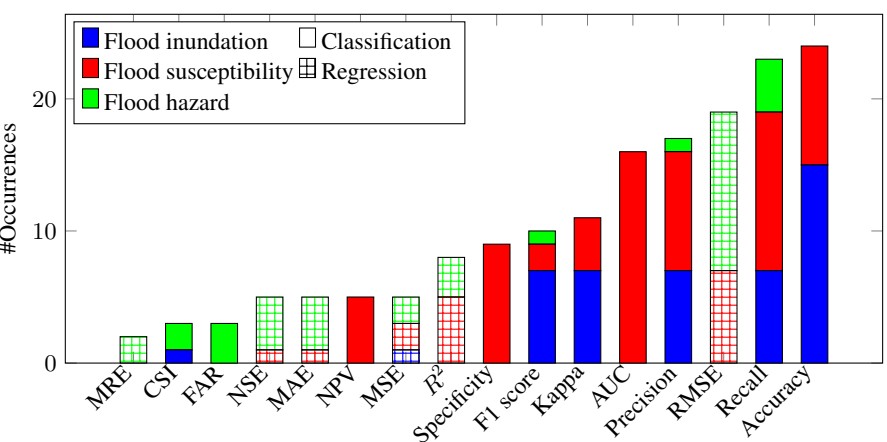

**Figure A1.** Distribution of the comparison metrics in the reviewed papers per type of application.

Abbreviations: AUC = Area Under the ROC Curve; CSI = Critical Success Index; FAR = False Alarm Ratio; MAE = Mean Average Error; MRE = Mean Relative Error; MSE = Mean Squared Error; NPV = Negative Predictive Value; NSE = Nash-Suitcliffe Efficiency; RMSE = Root Mean Squared Error; $R^2$ = Coefficient of determination

**Appendix B:  Flood susceptibility inputs**

Figure 1b shows the complete list of the inputs used in the flood susceptibility models. Inputs which were considered only once were discarded from this graph.





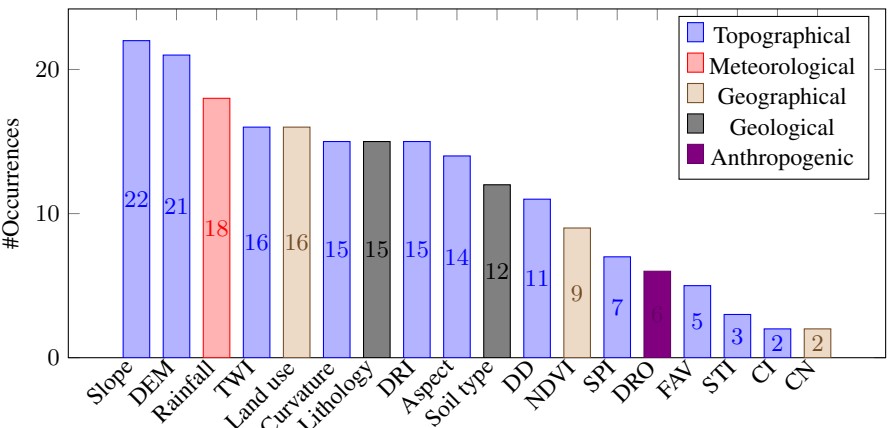

**Figure B1.** Distribution of the inputs for flood susceptibility for the 23 reviewed papers. The inputs are categorized in topographical, meteorological, geographical, geological, and anthropogenic factors. Inputs which were considered only once were discarded from this graph. Abbreviations: CI = Convergence Index; CN = Curve Number; DD = Drainage Density; DEM = Digital Elevation Model; DRI = Distance from Rivers; DRO = Distance from Roads; FAV = Flow Accumulation Value; NDVI = Normalized Difference Vegetation Index; SPI = Stream Power Index; STI = Sediment Transport Index; TWI = Topographic Wetness Index

## Appendix C: Reviewed papers

Table C1: A brief description of the reviewed papers, following the same ordering as in Table 2.

| Reference | Brief description |
|---|---|
| Li et al. (2015) | A MLP combined with genetic algorithm is used for super-resolution mapping of wetland inundation. Images are taken from Landsat remote sensing for two areas, in China and Australia. |
| Li et al. (2016a) | A MLP optimized with particle swarm optimization is used for super-resolution mapping. Images are taken from Landsat remote sensing for two areas, in China and Australia. |
| Nogueira et al. (2017) | Satellite image patches, collected from eight flooding events, are classified in flooded and non flooded areas using a CNN. |
| Gebrehiwot et al. (2019) | A set of images taken from unmanned aerial vehicles is used for flood inundation mapping using a CNN. The dataset is composed by three study areas in North Carolina, USA, with a total of 100 images. |
| Ichim and Popescu (2020) | A combination of five different CNN based models is used for flood inundation mapping. A set of 2000 images derived from UAVs in a Romanian rural region is classified into flooded, vegetation and non-flooded areas. |
| Hou et al. (2021) | A flood propagation experiment in a small scale laboratory is used to develop a CNN model for inundation mapping. The dataset is composed by a sequence of 1930 images detected from cameras. |
| Hashemi-Beni and Gebrehiwot (2021) | A encoder-decoder CNN is developed for flood inundation mapping. The study area is the same as Gebrehiwot et al. (2019). |
| Wieland and Martinis (2019) | A encoder-decoder CNN model is trained to identify five classes (water, land, snow, shadow, cloud) from Landsat and Sentinel-2 satellite images. |
| Kang et al. (2018) | A CNN model is developed for flood detection from satellite images for three study areas in China. |
| Sarker et al. (2019) | A CNN model is trained to predict flood extent from Landsat satellite images across Australia. |
| Nemni et al. (2020) | A CNN model is trained to predict flood extent from Sentinel-1 SAR imagery. The United Nations Satellite Centre (UN-OSAT) flood dataset, which covers flood events from Africa and East Asia, is used for the model development. |
| Isikdogan et al. (2017) | A CNN model is trained to identify five classes (water, land, snow, shadow, cloud) from Landsat satellite images. |
| Amini (2010) | High-resolution images are classified into five categories with a MLP. The study area is located in a Iranian city. |





| | |
|---|---|
| Li et al. (2016b) | A MLP is developed for super-resolution mapping. Images are taken from Landsat remote sensing for two areas, in China and Australia. |
| Peng et al. (2019) | A CNN which considers pre- and post-flood satellite images is used for urban flood detection. The study area is in Texas, USA, and considers two hurricane events. |
| Dong et al. (2021) | A combination of 1D-CNN and GRU is used for prediction of flood inundation in urban area in Texas, USA. The dataset is composed by a channel sensor network. Given information on the temporal evolution of water depth and precipitation, for each sensor, the model predicts which nodes will be flooded. |
| Liu et al. (2019) | A CNN which considers pre- and post-flood SAR images is used for coastal flood detection. The study area is in Texas, USA, and considers six pairs of satellite images for a hurricane event. |
| Muñoz et al. (2021) | A CNN model is developed for compound flood mapping for the Atlantic coast of USA. The model takes as input Landsat and SAR satellite data, along with a DEM of the study areas, and combines them with data fusion. The model distinguishes then between several categories, such as flooded areas and vegetation. |
| Syifa et al. (2019) | A MLP which considers pre- and post-flood satellite images is used for flood inundation mapping after a dam break in Brazil. |
| Khoirunisa et al. (2021) | A MLP is developed to obtain a flood susceptibility map for a Taiwanese city, using nine flood conditioning factors. The model is then compared with SOBEK, a 2D hydraulic model. The data is given by 307 flood events collected between 2015 and 2019. |
| Jahangir et al. (2019) | A MLP is employed in a river basin in Iran to predict discharges, using seven flood conditioning factors. The discharges are then converted into a flood susceptibility map via a GIS software. The data is given by hydrometric stations in the basin. |
| Ahmadlou et al. (2021) | Several MLP architectures are presented for the development of two flood susceptibility maps in Iran and India using, respectively, nine and twelve flood conditioning factors. The datasets are composed by 147 and 300 flood events, respectively. |
| Popa et al. (2019) | A combination of MLP with frequency ratio is used to determine flash flood and river flood potential indexes for a Romanian catchment, using respectively 14 and 13 flood conditioning factors. The datasets are composed respectively by 168 and 172 flood locations. |
| Kia et al. (2012) | A MLP is employed in a river basin in Malaysia to predict discharges, which are then converted into a flood susceptibility map via GIS. The model considers seven flood conditioning factors. |
| Ahmed et al. (2021) | Two MLP models are developed to obtain a flood susceptibility map for a catchment in Bangladesh, using 12 flood conditioning factors. The data is given by 521 flood events collected during 2019. |
| Chakrabortty et al. (2021b) | Two MLP models are developed to obtain a flood susceptibility map for a catchment in India, using 15 flood conditioning factors. The models are then used estimate future flood susceptibility scenarios by considering rainfall variations due to climate change. |
| Saeed et al. (2021) | A MLP model is developed to obtain a flood susceptibility map for a basin in Pakistan, using nine flood conditioning factors. |
| Wang et al. (2020b) | Three CNN models, based on 1D, 2D, and 3D convolutional layers are proposed for a region in China using 13 flood conditioning factors. The dataset is based on 108 historical flood events. The model is also compared with support vector machine. |
| Khosravi et al. (2020) | A national scale flood susceptibility map is developed for Iran, using a CNN. The model uses ten flood conditioning factors and a dataset composed by 2769 flood events. |
| Fang et al. (2020a) | A LSTM model is developed for flood susceptibility in a Chinese county. The model uses 11 flood conditioning factors and a dataset of 108 flood events. The model is also compared with MLP, 1D-CNN, and 3D-CNN. |
| Tien et al. (2020) | A MLP, based on nine flash flood conditioning factors, is developed for a province in Vietnam. The dataset contains 732 flood events. |
| Ngo et al. (2018) | A combination of MLP with firefly algorithm optimization technique is proposed for flash flood susceptibility in a Vietnamese region. The model considers 12 flood conditioning factors and a dataset of 654 flooded areas |
| Costache et al. (2020) | Two MLP models, combined with analytical hierarchy process and frequency ratio, respectively, are developed for flash flood susceptibility mapping in a catchment in Romania. The model considers ten flood conditioning factors and 178 flooded locations. |
| Chakrabortty et al. (2021a) | Two MLP models are developed to obtain a flash flood susceptibility map for a catchment in India, using 14 flood conditioning factors. |
| Kourgialas and Karatzas (2017) | A national-scale flash flood susceptibility map is obtained with a MLP for Greece. The model considers seven flood conditioning factors and a dataset of 600 flood events. |





| | |
|---|---|
| Panahi et al. (2021) | Two models, CNN and RNN, are developed for flash flood susceptibility in a province in Iran. The model considers nine flood conditioning factors and a dataset of 143 flood events. |
| Liu et al. (2021) | A 1D-CNN model combined with fuzzy membership value is proposed for flood susceptibility mapping for a basin in China, using nine flood conditioning factors. The dataset is based on 485 flood locations. The model is also compared with support vector machine. |
| Darabi et al. (2021) | A MLP ensemble model is used for urban flood susceptibility mapping in a Iranian city. The model uses six flood conditioning factors and a dataset of 118 flooded locations. |
| Kalantar et al. (2021) | Three MLP models, one of which optimized with particle swarm optimization, are developed for flood susceptibility mapping in a urban catchment in Australia. The model considers 13 flood conditioning factors and 128 flooded locations. |
| Zhao et al. (2020b) | Two CNN models are used to assess urban flood susceptibility for a Chiese catchment. The model considers nine flood conditioning factors and 202 flooded locations. The model is also compared with support vector machine and random forest. |
| Zhao et al. (2021b) | A pre-trained CNN model, taken from Zhao et al. (2020b), is deployed in two urban areas, with data-rich and data-scarce scenarios. |
| Lei et al. (2021) | Two models, CNN and RNN, are developed for urban flood susceptibility in Seoul, South Korea. The model considers 10 flood conditioning factors and a dataset of 295 flood events. |
| Chu et al. (2020) | A MLP is used to estimate the water depth for a river reach in Australia. The dataset is generated using a 2D hydrodynamic model, TUFLOW, and considers ten flood events. |
| Lin et al. (2020b) | A MLP is used to estimate the water depth and extent for a river reach in a German city. The dataset is generated using a 2D hydrodynamic model, HEC-RAS, and considers 180 flood events. |
| Lin et al. (2020a) | A MLP is used to forecast water depth and flood extent for a river reach in a German city. The dataset is the same as in Lin et al. (2020b). |
| Huang et al. (2021a) | Two models, MLP and RNN, are developed for water depth estimation for an American river. Before being given as input to the models, the data is clustered based on slope, drainage area and hydrologic length. The dataset is generated using a 2D hydrodynamic model and considers 20 flood events. |
| Xie et al. (2021) | Three MLPs are used to estimate the water depth for a river reach in Australia (also considered by Chu et al. (2020)). The dataset is generated using a 2D hydrodynamic model, TUFLOW, and considers nine flood events. |
| Jacquier et al. (2021) | A combination MLPs with reduced-order modelling is applied to a river and a dam break flood simulations. Moreover, the model provides uncertainty estimates using ensembles and Bayesian neural networks. The dataset is obtained using a 2D hydrodynamic model, CuteFlow. |
| Kabir et al. (2020) | A 1D-CNN model is used to predict a flood hazard map, from a discharge hydrograph, for a river reach in England. The dataset is generated using a 2D hydrodynamic model, LISFLOOD-FP, and considers 10 flood events. The model is also compared with support vector regression. |
| Hosseiny (2021) | A encoder-decoder CNN is used for flood mapping in a river reach in the USA. The dataset is generated using a 2D hydrodynamic model, iRIC, and considers 7 flood events. |
| Guo et al. (2021) | A encoder-decoder CNN is used for flood mapping in three catchments, two in Switzerland and one in Portugal. The dataset is generated using a cellular-automata flood model, CADDIES, and considers 18 flood events. |
| Zhou et al. (2021) | A LSTM model is used to predict the temporal evolution of water depth in few representative locations along an Australian river. The water depths are then interpolated to obtain a flood hazard map. The dataset is generated using a 2D hydrodynamic model, TUFLOW, and considers 74 flood events. |
| Kao et al. (2021) | A stacked encoder decoder LSTM is used to determine flood hazard maps in time from precipitation. The study area is located in Taiwan. The dataset is generated using a storm water management model, HEC-1, plus a 2d hydrodynamic model and considers 24 flood events. |
| Chang et al. (2010) | A MLP is developed to forecast one hour ahead flood hazard maps in a Taiwanese county. The data is also pre-processed by clustering. The dataset is generated using a storm water management model, HEC-1, plus a 2d hydrodynamic model and considers 120 flood events. |
| Yokoya et al. (2020) | A encoder-decoder CNN is used for flash flood and debris flow mapping in Japan. The dataset is generated using a 2D hydrodynamic model coupled with a debris-flow module and considers 160 flood events. The model is then trained to estimate flood depths and debris flows from pre- and post-flood event images. |
| Berkhahn et al. (2019) | A MLP is used to predict maximum water levels for two urban areas. The dataset is generated using a coupled sewer-surface model, HE 2D, and considers 64 flood events. |



| Löwe et al. (2021) | A encoder-decoder CNN is used for flood mapping in a urban area in Denmark. The dataset is generated using a 2D hydrodynamic model, MIKE 21, and considers 53 flood events. |
| Hu et al. (2019) | A LSTM model is developed to simulate a tsunami in Japan. The model is trained in a lower dimensional space to reduce the problem's complexity. The dataset is obtained using a 3D hydrodynamic model, Fluidity, and consists of 100 snapshots of a modelled tsunami event. |

830 *Author contributions.* All authors contributed in conceptualizing the paper and its contents. RB and RT developed the structure of the paper. RB wrote the paper, produced all figures and tables, and formatted the article. RT, EI and SNJ reviewed, revised, and supervised the progress of the paper.

*Competing interests.* No competing interests are present.

*Acknowledgements.* This work is supported by the TU Delft AI Labs programme.





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
