# Peer review of "Deep Learning Methods for Flood Mapping: A Review of Existing Applications and Future Research Directions"

_Hydrology and Earth System Sciences, 2022_

## Author Response (AR1)

**Reply to Reviewer #1**

>> We thank the Reviewer for following up with the review process and for the additional comments provided. We address each of them in the following lines.

> I think the authors have properly addressed my comments in the previous version. I wish to thank the authors for their efforts in producing a paper that will actually also be useful in the classroom. I suggest accepting the paper after the minor comments below have been addressed:

> line 380: please check this sentence, I think it is the other way round, MAE is more robust. See the small numerical experiments here: https://towardsdatascience.com/comparing-robustness-of-mae-mse-and-rmse-6d69da870828

>> We thank the Reviewer for pointing this out. We agree with this consideration so we modified lines 382-383 as follows: "In general, MAE may be preferred to RMSE since the latter is heavily influenced by the presence of extreme outliers."

> Section 3.3.3 - I think it would be good to clarify in this section and maybe also earlier when introducing the different applications, that flood inundation in this paper refers to mapping applications (in contrast to forecasting of flood inundation based on some meteorological input, which will yield much lower scores)

>> We added a sentence in section 3.3 to better clarify what we refer to as flood inundation maps: "We remind the reader that, in this paper, we refer to flood inundation as the process of mapping flooded and non-flooded areas from a picture of a flood"

>> Moreover, we added a link to such a definition in section 3.3.3 as follows: "As defined in section 3.3, flood inundation mapping determines which cells of the flood picture are represented as flooded or not."

> line 570: The logical flow does not make sense. You start outlining limitations of the methodological approaches and then suddenly jump to new applications. It's not clear how these are connected. I would say these text blocks belong into separate subsections

>> We thank the Reviewer for this comment. As the previous version was confusing, we redistributed the limitations of each methodological approach throughout the results section, when each application is addressed. In this way, section 4.1 now focuses directly on new applications where DL could be applied.

>> We show in the following paragraphs how the limitations have been redistributed for each application (in italic).

Lines 439-443: "Satellite data and UAV imagery are both remote sensing data that represent a flood event seen from above. The main differences concern the scale, the resolution, and the availability. UAVs are applicable only for small areas but their resolution is higher than satellite data. UAVs can be readily used but may be unavailable in certain areas. On the other hand, satellite data is available worldwide but its frequency of observation can be limiting. *Satellites can also struggle to extract information below clouded areas (e.g., Meraner et al., 2020).*"

Lines 495-499: "Topographical data were the most frequent type of input. Many papers present a sensitivity analysis to determine which factors influenced the most the final results: on average, these were slope, land use, aspect, terrain curvature, and distance from the rivers (e.g., Khosravi et al., 2020; Fang et al., 2020a; Popa et al., 2019; Costache et al., 2020). A complete list of inputs is reported in the Appendix (Fig. B1). *As there are several typologies of inputs, it is important to design an appropriate model to integrate heterogeneous environmental information.*"

> line 804 - 809: Maybe this can be formulated a bit more soft. Initial suggestions for accounting for changing terrain etc. have been published, but there certainly is more work to be done in this direction.

>> We understand the point of the Reviewer. Hence, we modified the paragraph in lines 809-815 as follows:

"Current deep learning models struggle to generalize across different case studies and regions, implying that a new model must be created each time. Further problems occur when modelling the complex interactions with the natural and built environment. While some of the reviewed papers provide initial suggestions to tackle these issues, the community should invest more efforts in this direction. A possible solution to these problems is to use novel DL architectures that include meshes as learning frameworks. Mesh-based neural networks, such as graph neural networks and neural operators, can consider arbitrarily shaped domains and thus provide the required flexibility to generalize across case studies and model the effects of complex interactions."

**Reply to Reviewer #2**

>> We thank the Reviewer for the comments. We address each of them in the following lines.

> This paper presents a review of the applications of deep learning models for flood inundation, susceptibility, and hazard mapping in a period between 2010 and 2021, leading to reviewing a total of 58 papers. The manuscript is well-rewritten and organized and provides information for international readers. However, two important weaknesses exist; these can both be strengthened relatively easily. (1) The objectives should be better presented. A numbered list of objectives would work better.

>> We thank the reviewer for this suggestion. We have organized the main contributions of our paper in a list, in lines 70-76:

"The main insights from this paper can be summarized as follows:

1. We identify common patterns and deduce general considerations based on the presented results while highlighting individual innovative approaches;
2. We compare against traditional methods to further validate the benefits of employing DL models;
3. We identify a series of current knowledge gaps and propose possible solutions to them drawing from recent advancements in DL."

> (2) The conclusions section should be rewritten and further improved to address the objectives. This section should also include the limitations of the current work and suggestions for further research in the future. For example, the authors can suggest a more systematic Bibliometric/scientometric analysis and the use of several well-established metrics for future research. See https://www.mdpi.com/2071-1050/13/15/8261/htm are the references therein.

>> We thank the Reviewer for the suggestion. We have now modified the conclusions section to enforce a clearer connection between the research objectives and the conclusions we obtained, in lines 775-805:

"This paper presented a review of current applications of deep learning models for flood mapping. The chosen search criteria yielded a total of 58 papers published between 2010 and 2021. From our analysis we found common patterns that can be summarized as follows:

[revised manuscript text omitted]